# Beige adipocytes mediate the neuroprotective and anti-inflammatory effects of subcutaneous fat in obese mice

De-Huang Guo[1], Masaki Yamamoto[1], Caterina M. Hernandez [2], Hesam Khodadadi[3], Babak Baban[3,4] & Alexis M. Stranahan [1✉]

Visceral obesity increases risk of cognitive decline in humans, but subcutaneous adiposity does not. Here, we report that beige adipocytes are indispensable for the neuroprotective and anti-inflammatory effects of subcutaneous fat. Mice lacking functional beige fat exhibit accelerated cognitive dysfunction and microglial activation with dietary obesity. Subcutaneous fat transplantation also protects against chronic obesity in wildtype mice via beige fat-dependent mechanisms. Beige adipocytes restore hippocampal synaptic plasticity following transplantation, and these effects require the anti-inflammatory cytokine interleukin-4 (IL4). After observing beige fat-mediated induction of IL4 in meningeal T-cells, we investigated the contributions of peripheral lymphocytes in donor fat. There was no sign of donor-derived lymphocyte trafficking between fat and brain, but recipient-derived lymphocytes were required for the effects of transplantation on cognition and microglial morphology. These findings indicate that beige adipocytes oppose obesity-induced cognitive impairment, with a potential role for IL4 in the relationship between beige fat and brain function.

[1] Department of Neuroscience and Regenerative Medicine, Medical College of Georgia, Augusta University, Augusta, GA, USA. [2] Department of Pharmacology and Toxicology, Medical College of Georgia, Augusta University, Augusta, GA, USA. [3] Department of Oral Biology, Medical College of Georgia, Augusta University, Augusta, GA, USA. [4] Plastic Surgery Section, Department of Surgery, Medical College of Georgia, Augusta University, Augusta, GA, USA. ✉email: astranahan@augusta.edu

The prevalence of obesity is increasing rapidly[1], and in the context of an aging population, the obesity epidemic has the potential to exacerbate projected increases in age-related cognitive decline and dementia[2,3]. Visceral adiposity at midlife predicts subsequent rates of dementia independently of weight loss during the intervening decades, suggestive of a critical window for dysfunction in the adult brain[4]. Fewer studies have focused on subcutaneous adiposity and cognitive decline, but cross-sectional data suggest that the "pear-shaped" distribution of body fat does not increase rates of mild cognitive impairment or dementia, and may be protective[5]. Vulnerability to age-related cognitive decline with visceral or subcutaneous adiposity, therefore, follows the same pattern as risk for metabolic complications[6], but mechanism(s) linking subcutaneous adiposity with memory have yet to be identified.

Unlike visceral fat, which contains a homogeneous population of white adipocytes, subcutaneous fat contains both white adipocytes and "beige" adipocytes that expend energy in a manner analogous to brown fat[7]. Beige adipocytes interact continuously with immune cells, and the acquisition of thermogenic features (beiging) requires induction of the anti-inflammatory cytokines interleukin-4 (IL4) by leukocytes in subcutaneous adipose tissue (SAT)[8,9]. Given the importance of IL4 signaling for allergic responses and autoimmunity, circulating concentrations of IL4 are tightly regulated[10,11]. However, tissue-resident immune cells are capable of efflux and migration to other organs, including the brain and leptomeninges[12,13]. Immune cell trafficking and local signaling at the blood–brain and blood–cerebrospinal fluid interfaces, therefore, enables inter-organ crosstalk independently of circulating factors or direct access to the brain parenchyma[12]. Although peripheral macrophages gain access to the brain with chronic obesity, less is known about earlier neuroimmune interactions at the blood–brain and blood–cerebrospinal fluid interfaces[14,15]. Moreover, despite emerging roles in other chronic inflammatory diseases[13], T-lymphocytes have received less attention with respect to their protective or pathogenic roles in the brain with obesity.

Here, we investigated immunoregulatory interactions between beige adipocytes and cognition through a series of dietary obesity and SAT transplantation experiments in male mice. The results indicate that beige adipocytes are indispensable for the neuroprotective and anti-inflammatory effects of subcutaneous fat, and implicate beige fat-stimulated IL4 production by meningeal lymphocytes in communication between SAT and the CNS.

## Results

**Beige adipocytes in subcutaneous fat oppose peripheral inflammation with dietary obesity.** Expression of the transcription factor PRDM16 is required for beiging in subcutaneous fat[7,16]. To investigate potential relationships between beige adipocytes and obesity-induced inflammation, transgenic mice lacking beige fat (Adiponectin$^{cre}$/Prdm16$^{fl/fl}$, Tg) and non-transgenic littermate controls (nTg) were maintained on a low-fat or high-fat diet (LFD, HFD) for 1 month. Weight gain was similar in Tg and nTg mice over 4 wk (Fig. 1a), and the proportional weights of SAT, epididymal visceral adipose tissue (VAT), and interscapular brown adipose tissue (BAT) did not differ between genotypes (Fig. 1b). Beiging is accompanied by induction of thermogenic genes and formation of UCP1$^+$ multilocular adipocytes[7]. These signatures are increased by cold exposure, but are also present at room temperature, which falls below thermoneutrality in mice[16,17]. Anatomically, beige adipocytes are highly enriched in inguinal SAT, relative to the dorsolumbar region[17]. To control for this anatomical gradient, tissue punches from paraffin-embedded inguinal SAT were used for gene

expression endpoints after HNE staining on adjacent sections. qPCR analysis of thermogenic (*Ucp1* and *Ppargc1a*) and brown fat differentiation genes (*Cidea*) revealed reduced expression in Tg mice, relative to nTg/LFD samples (Supplementary Fig. 1A). Reductions were comparable in Tg/LFD and Tg/HFD samples, and there was no loss of beiging-associated gene expression in samples from nTg/HFD mice (Supplementary Fig. 1A). Expression of *Prdm16* was significantly reduced in Tg mice, irrespective of diet (Supplementary Fig. 1A), consistent with the original report characterizing this model[16]. Downregulation of thermogenic genes was not attributable to global reductions in gene expression, as the adipocyte differentiation gene *Pparg* was similarly elevated in SAT from nTg/HFD mice and Tg/HFD mice (Supplementary Fig. 1A).

To examine histological signatures of beiging, we analyzed multilocular adipocytes on HNE-stained sections from inguinal SAT (Supplementary Fig. 1B). Blinded visual inspection of (500–700) systematic random sampling fields from each animal revealed that multilocular adipocytes were less frequent in Tg mice, irrespective of diet (Supplementary Fig. 1C, D; multilocular adipocytes [% of fields], mean ± sem, $n = 4$/condition: nTg/LFD = 32.0 ± 2.9; nTg/HFD = 30.6 ± 2.5; Tg/LFD = 10.5 ± 1.6; Tg/HFD = 10.6 ± 0.9). Using the same approach, we observed that UCP1$^+$ multilocular adipocytes were less prevalent in inguinal SAT from Tg mice on either diet (Supplementary Fig. 1E, F; UCP1$^+$ multilocular adipocytes [%], mean ± sem, $n = 4$/condition: nTg/LFD = 17.1 ± 1.9; nTg/HFD = 17.8 ± 1.7; Tg/LFD = 7.5 ± 1.2; Tg/HFD = 6.3 ± 1.5). Inguinal SAT receives greater sympathetic innervation than dorsolumbar SAT in normal-weight Wt mice, and sympathetic activity promotes beiging[7,17]. For qualitative insight into innervation, we performed immunohistochemistry for tyrosine hydroxylase (TH) in inguinal SAT. TH-stained fiber bundles were observed along the surface of inguinal SAT, with penetrating fibers interdigitating between individual adipocytes (Supplementary Fig. 1E). There was no overt effect of diet or genotype on the distribution of TH-labeled fibers (Supplementary Fig. 1E), consistent with the integrity of sympathetic innervation reported using 3-dimensional methods[17,18]. Interscapular BAT also appeared normal, with no obvious differences in multilocular adipocytes or UCP1 immunoreactivity (Supplementary Fig. 2A, B), as reported previously[16]. Multilocular adipocytes were not observed in VAT (Supplementary Fig. 2C), and UCP1 immunoreactivity was rare (Supplementary Fig. 2D). Taken together, these patterns are consistent with selective loss of beige adipocytes in SAT from Adiponectin$^{cre}$/Prdm16$^{fl/fl}$ mice.

Inflammation in obesity promotes metabolic pathology and contributes to obesity-induced cognitive dysfunction[6,15]. While there was no effect of genotype or 4 wk HFD consumption on glucose tolerance (Fig. 1c), analysis of serum cytokines revealed significantly greater upregulation of interleukin-1beta (IL1b) in Tg/HFD, relative to nTg/HFD (Fig. 1d). One month HFD consumption also increased circulating levels of TNFa, and this effect was comparable between genotypes (Fig. 1d). Ablation of beige adipocytes in SAT converts the tissue environment to a pro-inflammatory state reminiscent of visceral fat, based on induction of visceral-selective genes and increased macrophage infiltration with dietary obesity in Adiponectin$^{cre}$/Prdm16$^{fl/fl}$ mice with high-fat feeding[16]. To determine whether similar changes occurred under the current experimental protocol, we quantified pro-inflammatory cytokine gene expression and visualized F4/80$^+$ crown-like structures in SAT and VAT from Adiponectin$^{cre}$/Prdm16$^{fl/fl}$ mice and nTg littermates on HFD or LFD (Supplemental Fig. 3A). Crown-like structures were more prevalent in SAT from Tg/HFD mice, relative to nTg/HFD mice (Supplemental Fig. 3A; % fields with crown-like structures, mean ± sem, $n = 4$ mice/condition: nTg/LFD = 1.8 ± 0.3; nTg/HFD = 2.2 ± 0.3; Tg/LFD = 1.9 ± 0.6; Tg/HFD = 7.3 ± 1.2), and qPCR analysis

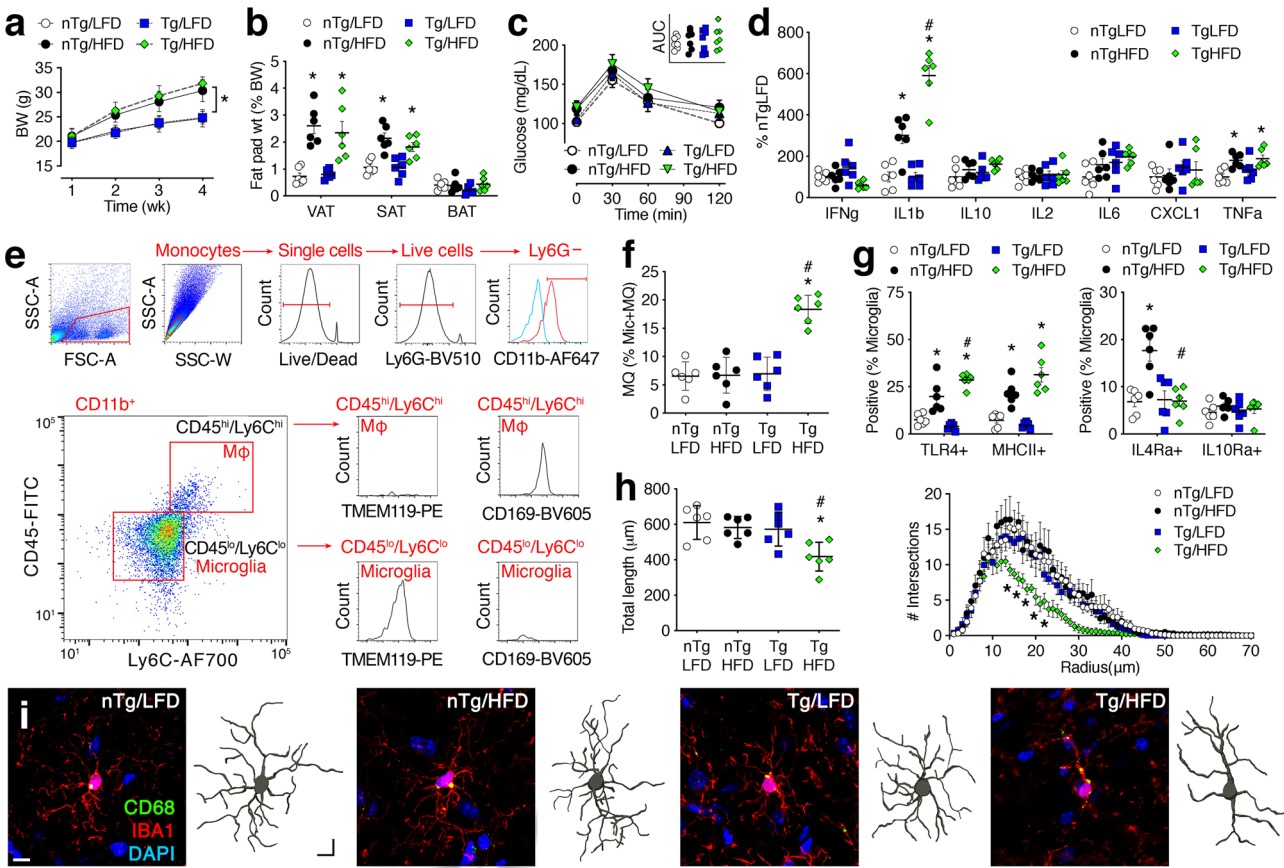

**Fig. 1 Transgenic mice lacking beige fat exhibit accelerated neuroinflammation with dietary obesity. a** Adiponectin$^{cre}$/PRDM16$^{fl/fl}$ mice (Tg) and nontransgenic (nTg) littermates gain similar amounts of weight on a low-fat or high-fat diet (LFD, HFD; symbols show group mean ± sem, $n = 24$/condition). **b** Weight of visceral, subcutaneous, and brown adipose tissue (VAT, SAT, BAT) as a percent of body weight (symbols represent individual mice, line shows mean ± sem, $n = 6$ mice/condition). **c** Intraperitoneal glucose tolerance testing revealed no group differences (symbols show group mean ± sem, $n = 7$ mice/condition). Area under the curve (AUC) (inset; symbols represent individual mice). **d** Serum pro-inflammatory cytokines ($n = 6$ mice/condition). Symbols represent individual mice and the line shows mean ± sem (applies to **d–f**). **e** Gating for analysis of microglia and macrophages (MΦ). Parent gate indicated in red text above each panel. **f** Accumulation of CD11b$^+$/CD45$^{hi}$/Ly6C$^{hi}$/TMEM119$^-$/CD169$^+$ macrophages (MQ) in the brains of Tg/HFD mice ($n = 6$ mice/condition). **g** Resident microglia from mice lacking beige fat exhibit greater induction of TLR4 and MHCII on HFD (left; $n = 6$ mice/condition). Beige adipocytes are required for co-induction of IL4Ra after 1-month HFD consumption (right; $n = 6$ mice/condition). **h** Hippocampal microglia exhibit reductions in process length (left) and complexity (right) in Tg/HFD mice. For length (left), symbols represent data from individual mice (average of 5 cells/mouse) and the line shows mean ± sem ($n = 6$ mice/condition). The same sampling scheme was applied for process complexity (right); symbols show group mean ± sem. **i** Immunofluorescence labeling for IBA1 and CD68 with a corresponding 3D reconstruction of IBA1$^+$ cells. Scale bar (10 μm) shown with nTg/LFD applies to all panels. For all graphs, white circles represent nTg/LFD; black circles, nTg/HFD; blue squares, Tg/LFD; green diamonds, Tg/HFD. Data in **a** were analyzed using 2-way repeated-measures ANOVA; for (B-H), 2-way ANOVA. *$p < 0.05$ relative to nTg/LFD; #$p < 0.05$ relative to nTg/HFD, determined by Tukey's multiple comparison test. For data, statistics, and exact $p$-values, see Source Data File 1.

revealed significant increases in *Ccl2*, *Il1b*, and *Tnfa* expression (Supplemental Fig. 3D). Accumulation of crown-like structures was comparable in VAT from nTg/HFD and Tg/HFD mice (Supplemental Fig. 3C; mean ± sem, $n = 4$/condition: nTg/LFD = 1.8 ± 0.5; nTg/HFD = 8.6 ± 1.0; Tg/LFD = 2.2 ± 0.4; Tg/HFD = 9.1 ± 1.1), as was induction of pro-inflammatory cytokines (Supplemental Fig. 3D). Overall, these data are consistent with the acquisition of VAT-like features in SAT from mice lacking beige fat after HFD consumption.

**Loss of beige adipose skews microglia toward pro-inflammatory activation with dietary obesity.** To determine whether loss of beige adipocytes alters obesity-induced inflammation in the brain, we isolated forebrain mononuclear cells (FMCs) from Adiponectin$^{cre}$/Prdm16$^{fl/fl}$ mice and nTg littermates after 4 wk on HFD or LFD. The phenotype and activation state of FMCs was determined using flow cytometry after gating on size and granularity, followed by doublet discrimination and dead cell exclusion (Fig. 1e). Monocytes were gated as Ly6G$^-$/CD11b$^+$ and peripheral macrophages were gated as CD45hi/Ly6Chi monocytes, as shown (Fig. 1e). Intensity-based disambiguation of resident microglia and infiltrating macrophages is complicated by the upregulation of phenotypic antigens in chronic inflammatory diseases[19]. We therefore included CD169 as an additional marker of infiltrating macrophages and TMEM119 as a marker of resident microglia[20,21].

These experiments revealed increases in obesity-induced microglial activation and macrophage infiltration in mice lacking beige fat (Fig. 1f, g). The CD45$^{HI}$/Ly6C$^{HI}$/CD169$^+$/TMEM119$^-$ population was significantly larger in Tg/HFD mice, relative to all other groups (Fig. 1f). This effect was specific to monocytes, as there were no group differences in the proportion of Ly6G$^+$/CD11b$^-$ neutrophils in these experiments (% of live cells, mean ± sem, $n = 6$/condition: nTg/LFD = 0.35 ± 0.06; nTg/HFD = 0.41 ± 0.12;

Tg/LFD = 0.31 ± 0.08; Tg/HFD = 0.39 ± 0.07). Macrophage infiltration was accompanied by pro-inflammatory activation of resident microglia, as determined by subsequent analysis of TLR4 and MHCII in CD45$^{LO}$/Ly6C$^{LO}$/CD169$^{-}$/TMEM119$^{+}$ cells. Microglial induction of TLR4 was evident in nTg/HFD mice, relative to nTg/LFD mice, but these effects were amplified in Tg/HFD mice (Fig. 1g). Similar trends were observed for MHCII (Fig. 1g). Beige adipocytes in SAT interact with local immune cells, and these interactions are mediated in part by Th2 cytokines, such as IL4[8,9]. Stimulation with IL4 upregulates microglial IL4 receptor-alpha (IL4Ra)[22]. To examine whether early obesity promotes Th2 cytokine signaling in microglia, we quantified IL4Ra and interleukin-10 receptor-alpha (IL10Ra) expression in CD45$^{LO}$/Ly6C$^{LO}$/CD169$^{-}$/TMEM119$^{+}$ microglia. These data revealed increases in microglial IL4Ra expression in nTg/HFD, but not in Tg/HFD mice (Fig. 1g). There was no effect of diet or genotype on IL10Ra (Fig. 1g), but the stepwise induction of TLR4 and MHCII in microglia from nTg/HFD and Tg/HFD mice suggests that co-induction of IL4Ra might limit pro-inflammatory activation in obesity.

After observing a potential role for beige adipocytes in microglial polarization by flow cytometry, we quantified morphological and cellular indices of activation among IBA1$^{+}$ cells in the hippocampal dentate molecular layer. At the 4 wk time point, microglia from nTg/HFD mice did not differ from nTg/LFD based on total process length (Fig. 1h, left) or Sholl analysis of process complexity (Fig. 1h, right). By contrast, IBA1-labeled microglia from Tg/HFD mice exhibited reductions in total length (Fig. 1h, left) and process complexity (Fig. 1h, right). Changes in microglial ramification in Tg/HFD mice were accompanied by accumulation of CD68$^{+}$ puncta within microglial processes (Fig. 1i). There were no differences between nTg/LFD and Tg/LFD microglia, indicative of vulnerability to obesity-induced microglial activation in mice lacking beige adipocytes.

**Susceptibility to obesity-induced hippocampal dysfunction in mice lacking beige fat.** To investigate potential changes in cognition in mice lacking beige fat, Adiponectin$^{cre}$/Prdm16$^{fl/fl}$ (Tg) mice and nTg littermates were maintained on LFD or HFD for 4 wk, as shown (Fig. 1a). Non-overlapping cohorts of mice were tested in recognition memory paradigms before testing in the water maze or the Barnes maze, a spatial memory paradigm in which mice learn the location of an escape box located under one of twelve holes around the edge of a circular platform[23]. In the novel object preference task, Tg/HFD mice spent less time exploring the novel object, indicative of impaired memory for the familiar object presented 30 min earlier (Fig. 2a). This pattern was not explained by differences in total object exploration (% time with both objects, mean ± sem, n = 12/condition: nTg/LFD = 25.1 ± 2.3; nTg/HFD = 23.3 ± 2.5; Tg/LFD = 26.8 ± 1.8; Tg/HFD = 21.9 ± 2.5). Novel object preference testing was carried out after video tracking of exploratory behavior in an empty arena, as described[15]. There were no differences in total locomotor exploration (total distance in meters, mean ± sem, n = 12/condition: nTg/LFD = 0.93 ± 0.08; nTg/HFD = 0.83 ± 0.07; Tg/LFD = 1.01 ± 0.1; Tg/HFD = 0.84 ± 0.07) or exploration of the center in the empty arena (% in center, mean ± sem, n = 12/condition: nTg/LFD = 14.6 ± 1.12; nTg/HFD = 14.74 ± 0.81; Tg/LFD = 15.98 ± 0.91; Tg/HFD = 15.20 ± 0.63), suggesting that group differences in anxiety were unlikely in this setting. In the Y-maze, Tg/HFD mice alternated less frequently than nTg/HFD mice (Fig. 2b; for source data and p-values, see Source Data File 2). Testing was not complete until the mouse made seven arm entries, and there was no effect of diet or genotype on locomotor speed (cm/s, mean ± sem, n = 12/condition: nTg/LFD = 2.32 ± 0.19; nTg/HFD = 2.23 ± 0.10; Tg/LFD = 2.31 ± 0.19; Tg/HFD = 2.19 ± 0.10). Given the absence of group differences in locomotor speed under this entries-to-criterion

paradigm, reduced alternation in Tg/HFD mice likely reflects deficits in spatial recognition memory.

Subsequent analysis of hippocampus-dependent memory in the water maze revealed deficits in spatial memory acquisition in Tg/HFD mice (Fig. 2c, d). Tg/HFD mice exhibited significantly longer path lengths during hidden platform training, relative to Tg/LFD mice (Fig. 2c). Deficits in spatial memory were also evident in the probe trial, where Tg/HFD mice spent significantly less time swimming in the target quadrant, relative to all other conditions (Fig. 2d). There was no effect of genotype in mice on LFD, and all mice performed comparably when swimming toward a visible platform, indicative of intact visuomotor capabilities (swim distance in meters, mean ± sem, n = 12/condition: nTg/LFD = 7.3 ± 0.8; nTg/HFD = 7.2 ± 0.7; Tg/LFD = 6.3 ± 0.7; Tg/HFD = 6.5 ± 0.9).

Loss of beige fat impairs adaptive thermogenesis in SAT, which could alter physiological responses during training in the water maze[16,24]. To evaluate spatial learning and memory under less stressful conditions, an additional non-overlapping cohort of Tg and nTg mice were tested in the Barnes maze. Over the 4d training period, nTg/LFD mice navigated more efficiently to the escape hole, as indicated by reductions in distance traveled (Fig. 2e). One month of HFD consumption did not alter performance in nTg mice, but Tg/HFD mice exhibited significantly longer path lengths than all other groups (Fig. 2e; $F_{1,22}$ = 15.58, p < 0.01). Tg/HFD mice also exhibited less frequent nose pokes over the target hole during the probe trial (Fig. 2f). There was no effect of genotype in LFD mice, as Tg/LFD and nTg/LFD animals exhibited similar learning curves and probe trial performance (Fig. 2e, f). All groups of mice were able to find the escape box when its location was indicated by a proximal visual cue (distance [m], mean ± sem, n = 12/condition: nTg/LFD = 4.7 ± 0.7; nTg/HFD = 4.9 ± 0.9; Tg/LFD = 4.1 ± 0.5; Tg/HFD = 5.1 ± 0.7). Taken together, the behavioral results indicate that mice lacking functional beige fat are more vulnerable to obesity-induced cognitive deficits.

After observing increased susceptibility to obesity-induced microglial activation and cognitive impairment in mice lacking beige fat, we carried out electrophysiological recordings of long-term potentiation (LTP) at medial perforant path synapses on dentate gyrus granule neurons. These experiments revealed intact LTP in nTg mice after 4 wk HFD consumption (Fig. 2g). LTP deficits were evident in slices from Tg/HFD mice, based on comparisons of dendritic field potentials 50–60 min after high-frequency stimulation (Fig. 2g). To examine the role of local inflammation in these effects, slices were pre-incubated with the antibiotic minocycline (20 μM; Fig. 2h), which suppresses microglial M1 polarization without altering anti-inflammatory M2 activation in vivo and in vitro[25]. In the presence of minocycline, which was bath-applied throughout recording, slices from Tg/HFD mice exhibited normal LTP and did not differ from nTg/LFD controls (Fig. 2h, i). Collectively, these results implicate beige adipocytes in SAT as a source of signals that oppose obesity-induced neuroinflammation and cognitive impairment.

**Subcutaneous fat transplantation rescues hippocampal function in mice with dietary obesity.** We next considered whether increases in subcutaneous fat might offset the deleterious effects of obesogenic diets on memory and cognition in wild-type mice. To test this hypothesis and examine the potential role of beige adipocytes, male C57Bl6J mice were maintained on HFD for 10 wk before SAT transplantation from a lean nTg (TRANS$_{nTg}$) or Adiponectin$^{cre}$/Prdm16$^{fl/fl}$ Tg donor (TRANS$_{Tg}$; Fig. 3a). Parallel groups of mice were maintained on HFD or LFD before receiving

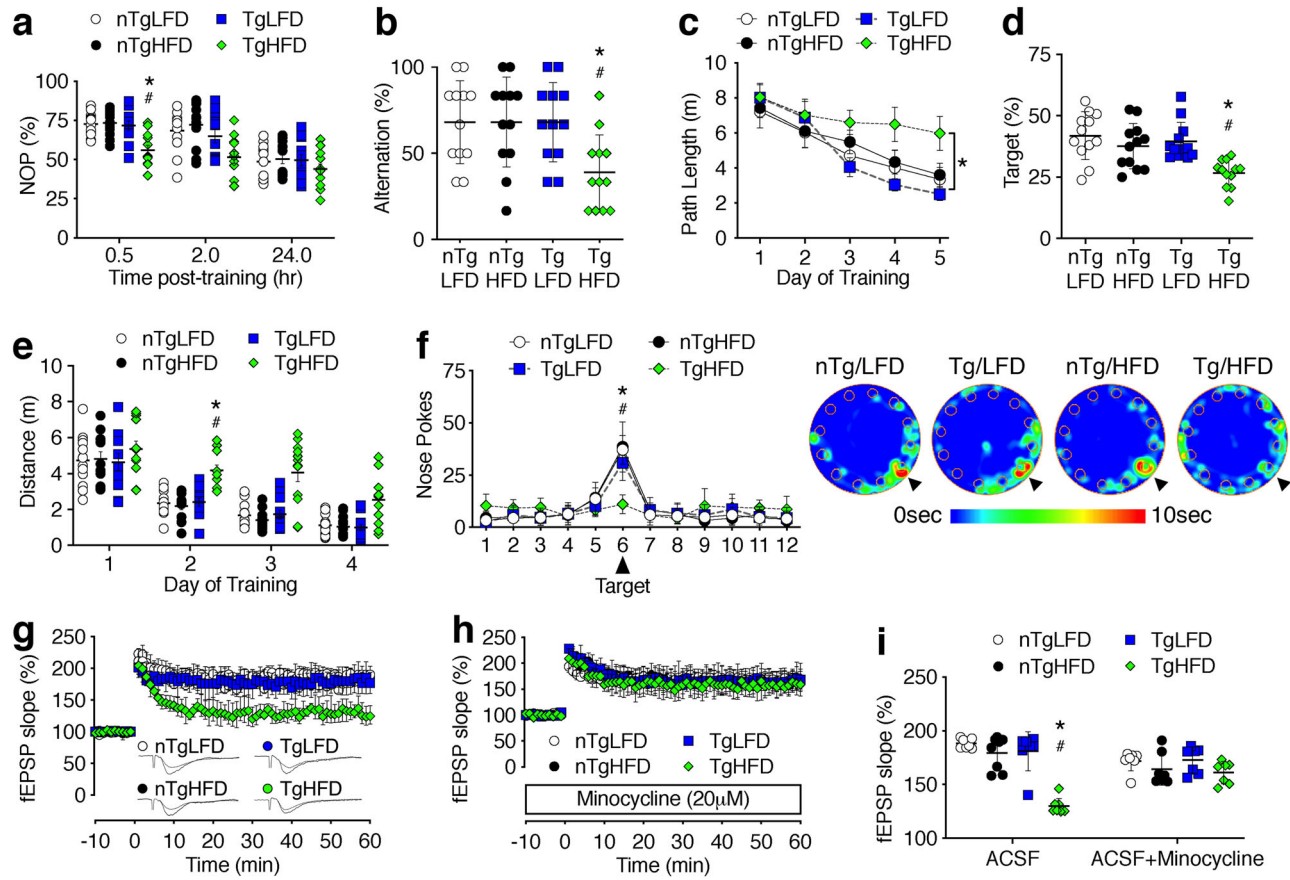

**Fig. 2 Early-onset hippocampal dysfunction with dietary obesity in mice lacking beige fat. a** Adiponectin$^{cre}$/PRDM16$^{fl/fl}$ mice (Tg) exhibit deficits in novel object preference (NOP) after 4 wk on a high-fat diet (HFD; $n = 12$ mice/condition). Symbol height shows data from individual mice and the line shows mean ± sem (applies to **a**, **b**). **b** Mice lacking beige fat exhibit reduced spatial recognition memory in the Y-maze ($n = 12$ mice/condition). **c**, **d** Impaired performance during water maze acquisition training (**c**) and probe testing (**d**) in Tg/HFD mice. For **c**, symbols represent mean ± sem; for **d**, symbols represent individual mice and the line shows mean ± sem ($n = 12$ mice/condition). **e**, **f** Deficits in spatial memory acquisition in the Barnes maze (**e**) and inaccurate search behavior during the probe trial (**f**) in Tg/HFD mice. For **e**, symbols represent individual mice and line shows mean ± sem; for **f** symbols represent group mean ± sem ($n = 12$ mice/condition). Heatmaps (**f**, right) show search patterns during probe trial for the indicated groups. **g** Tg/HFD mice exhibit early-onset deficits in hippocampal long-term potentiation (for nTg/LFD and Tg/LFD, $n = 7$ slice recordings from 3 mice; nTg/HFD, $n = 8$ slice recordings from 4 mice; Tg/HFD, $n = 8$ slice recordings from 5 mice). Symbols show mean ± sem. **h** Deficits were eliminated in the presence of minocycline (for Tg/HFD, $n = 7$ slice recordings from 4 mice; all other groups, $n = 7$ slice recordings from 3 mice). Symbols show mean ± sem. **i** Group differences in field excitatory postsynaptic potential (fEPSP) slope 50–60 min after induction (symbols represent individual slice recordings and line shows mean ± sem). For all graphs, white circles represent nTg/LFD; black circles, nTg/HFD; blue squares, Tg/LFD; green diamonds, Tg/HFD. For **a**, **c**, **e**, 2-way repeated-measures ANOVA; for **b**, **d**, **i**, 2-way ANOVA; *$p < 0.05$ relative to nTg/LFD; #$p < 0.05$ relative to nTg/HFD, determined by Tukey's multiple comparison test or Dunnett's T3 for heterogeneity of variance. For data, statistics, and exact $p$-values, see Source Data File 2.

sham surgery (HFD/SHAM, LFD/SHAM). Consumption of HFD increased body weights, but there was no effect of SAT transplantation on weight gain 2 wk after surgery (Fig. 3a). Dietary obesity also increased the proportional weights of epididymal VAT and resident SAT, but there was no effect of SAT transplantation on these parameters (Fig. 3b). Histological visualization of transplanted SAT revealed no relationship between donor genotype and rejection rates ($n = 4/48$ rejections in TRANS$_{nTg}$; $n = 5/50$ rejections in TRANS$_{Tg}$; Supplementary Fig. 4A, B).

Intrinsic features of SAT and extrinsic features of the subcutaneous environment contribute to depot-specific adipose tissue inflammation in obesity[6,7]. To determine whether beige adipocytes persist after transplantation into the peritoneal cavity, we visualized anatomical and gene expression signatures in the transplants. Multilocular adipocytes and UCP1 immunoreactivity were both present in transplants from nTg, but not Tg donors (Supplementary Fig. 5A, B), similar to the pattern observed in resident SAT from intact nTg and Tg mice (Supplementary

Fig. 1E–H). Transplantation did not restore gene expression for *Prdm16*, which was reduced in transplants from Tg donors (Supplementary Fig. 5D). Transplanted SAT from lean Tg donors also had lower levels of thermogenic (*Ucp1*, *Ppargc1a*) and brown/beige adipocyte differentiation gene expression (*Cidea*; Supplementary Fig. 5D). However, expression of the adipocyte differentiation gene *Pparg* was unaffected by donor genotype (Supplementary Fig. 5D), consistent with the similarities in resident SAT from lean nTg and Tg mice (Supplementary Fig. 1A).

We next evaluated the impact of SAT transplantation on glucose metabolism and circulating inflammatory cytokines. Intraperitoneal glucose tolerance testing 2 wk post-surgery revealed no significant differences in glycemic control (Fig. 3c). Multiplex profiling of serum cytokines revealed significant increases in IL1b and IL2 in HFD/SHAM samples, relative to LFD/SHAM (Fig. 3d). Surgical increases in subcutaneous fat reduced circulating IL2 concentrations, but had no effect on IL1b (Fig. 3d). Circulating CXCL1 concentrations were numerically

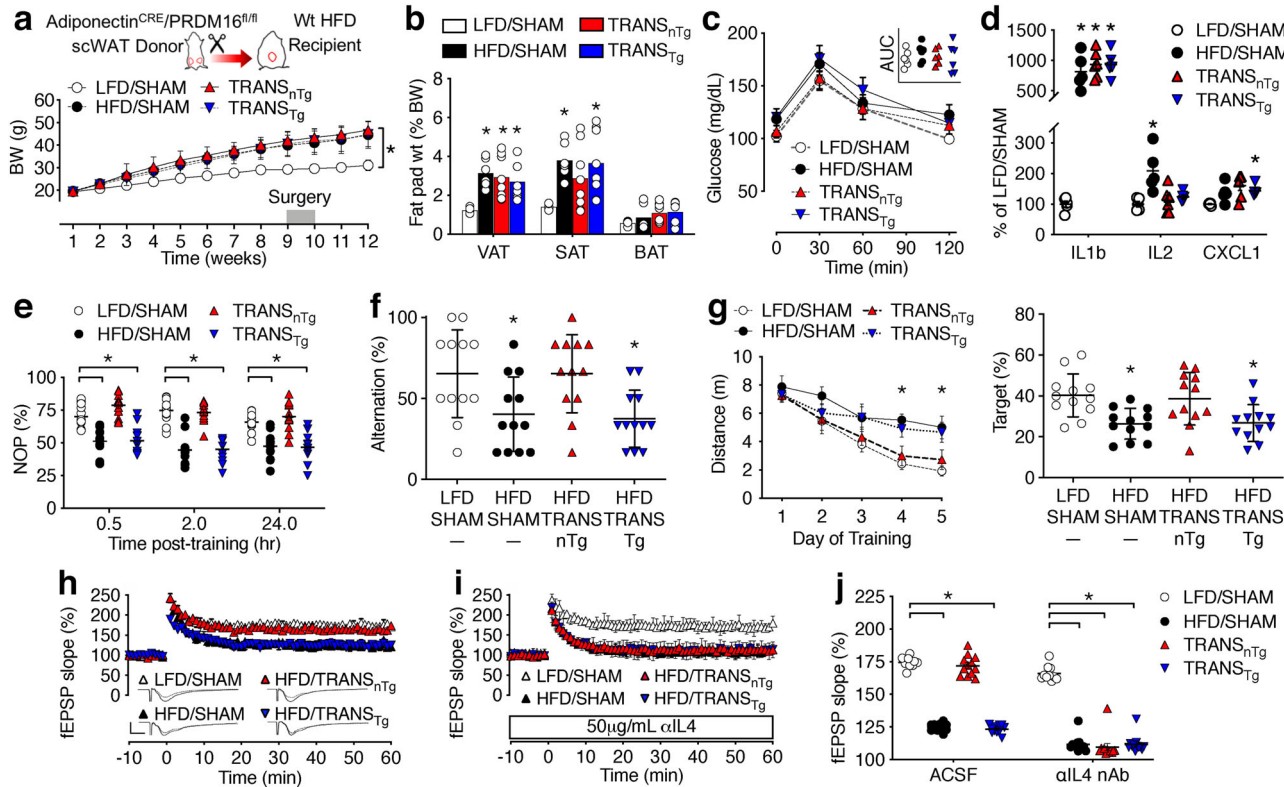

**Fig. 3 Subcutaneous fat transplantation rescues hippocampal function in obese mice via beige fat-dependent mechanisms. a** Weight gain in Wt mice maintained on a low-fat or high-fat diet (LFD, HFD), before and after sham operation (SHAM) or transplantation of subcutaneous fat from a nontransgenic (TRANS$_{nTg}$) or Adiponectin$^{cre}$/PRDM16$^{fl/fl}$ transgenic donor (TRANS$_{Tg}$). Symbols show group mean ± sem ($n = 24$ mice/condition). **b** Comparable increases in the proportional weight of visceral and subcutaneous adipose tissue (VAT, SAT) relative to body weight (BW) in HFD/SHAM ($n = 6$), TRANS$_{nTg}$ ($n = 8$), and TRANS$_{Tg}$ mice ($n = 8$), relative to LFD/SHAM ($n = 6$); no change in brown adipose tissue (BAT). Bar height shows mean, error bars show sem, and symbols represent individual mice. **c** No significant differences in glycemic control, based on area under the curve (AUC, inset). Symbols represent mean ± sem of ($n = 7$) mice/condition. For AUC (inset), symbols represent individual mice. **d** Dietary obesity increases serum pro-inflammatory cytokines (for TRANS$_{nTg}$, $n = 6$; all other groups, $n = 5$). Symbols represent individual mice and the line shows mean ± sem. **e** SAT transplantation restores novel object preference (NOP) via beige adipose-dependent mechanisms. Symbols represent individual mice and line shows mean ± sem ($n = 12$ mice/ condition). **f** Beige adipocytes are required for reinstatement of spatial recognition memory in the Y-maze after SAT transplantation ($n = 12$). **g** Surgical increases in subcutaneous fat rescue spatial memory acquisition (left; symbols represent group mean) and probe trial performance (right; symbols represent individual mice) in a beige fat-dependent manner. For both graphs, error bars show sem from ($n = 12$) mice/condition. **h** Beige adipocytes rescue dentate gyrus long-term potentiation (LTP) following SAT transplantation. For LFD/SHAM, symbols show mean ± sem of $n = 10$ slice recordings from 5 mice; for all other groups, symbols show the mean ± sem of $n = 12$ slice recordings from 6 mice (applies to **h**, **i**). **i** Preincubation with an IL4 neutralizing antibody (aIL4 nAb) blocks the effects of SAT transplantation on LTP. **j** Percent change in the dendritic field excitatory postsynaptic potential (fEPSP) during minutes 50–60 of recording (symbols represent individual recordings). For all graphs, white circles represent LFD/SHAM; black circles, HFD/ SHAM; red triangles, TRANS$_{nTg}$; inverted blue triangles, TRANS$_{Tg}$. For **a**, **c**, **e**, **g** [left], repeated-measures ANOVA; for **b**, **d**, **f**, **g** [right], **j**, one-way ANOVA. *$p < 0.05$, Tukey's multiple comparison test or Dunnett's T3 for heterogeneity of variance. For data, statistics, and exact $p$-values, see Source Data File 3.

higher in HFD mice, and this trend was statistically significant in TRANS$_{Tg}$ samples, relative to LFD/SHAM (Fig. 3d).

After investigating the consequences of acute SAT transplantation for systemic inflammation and metabolism, additional groups of Wt mice were maintained on LFD or HFD for 12 wk as shown (Fig. 3a), with cognitive testing 2 wk after SAT transplantation or sham surgery. In the object recognition paradigm, SAT transplantation restored novel object preference to levels that were identical to LFD/SHAM (Fig. 3e). Reinstatement of object recognition memory occurred in TRANS$_{nTg}$, but not TRANS$_{Tg}$ mice (Fig. 3e), suggestive of a requirement for beige adipocytes in the donor. There were no group differences in total locomotion (distance [m], mean ± sem, $n = 12$/ condition: LFD/SHAM = 0.96 ± 0.06; HFD/SHAM = 0.69 ± 0.08; TRANS$_{nTg}$ = 1.07 ± 0.12; TRANS$_{Tg}$ = 0.80 ± 0.11), or exploration in the center of the empty arena (% distance in center, mean ± sem, $n = 12$/condition: LFD/SHAM = 16.2 ± 1.2; HFD/SHAM = 12.6 ± 1.6; TRANS$_{nTg}$ = 15.2 ± 1.5; TRANS$_{Tg}$ = 17.8 ± 1.4). Spatial alternation in the Y-maze was also restored following SAT transplantation via

beige adipose-dependent mechanisms (Fig. 3f). In the water maze, HFD/SHAM and TRANS$_{Tg}$ mice exhibited significantly longer path lengths than LFD/SHAM during hidden platform training, indicative of deficits in spatial memory acquisition (Fig. 3g, left). Deficits in spatial memory acquisition were associated with impaired probe trial performance, as HFD/SHAM and TRANS$_{Tg}$ mice spent less time searching in the target quadrant than LFD/SHAM mice (Fig. 3g, right). By contrast, TRANS$_{nTg}$ mice did not differ from LFD/SHAM during acquisition training or in the probe trial (Fig. 3g). There were no group differences during visible platform training (distance [m], mean ± sem, $n = 12$/condition: LFD/SHAM = 5.7 ± 0.7; HFD/SHAM = 5.6 ± 0.5; TRANS$_{nTg}$ = 5.9 ± 0.7; TRANS$_{Tg}$ = 6.1 ± 0.8), consistent with beige adipose-mediated restoration of memory and cognition following SAT transplantation.

**SAT transplantation rescues hippocampal synaptic plasticity via IL4.** To investigate the synaptic mechanisms for beige

adipocyte regulation of learning and memory, we next carried out electrophysiological recordings in brain slice preparations. Consumption of HFD for 12 wk impaired LTP in sham-operated mice, as indicated by smaller increases in the slope of the dendritic fEPSP 1 h after high-frequency stimulation in slices from HFD/SHAM, relative to LFD/SHAM (Fig. 3h). SAT transplantation rescued LTP, but only when transplants were from an nTg donor (Fig. 3h). These effects likely represent postsynaptic responses, as presynaptic paired-pulse plasticity was unaffected (Supplementary Fig. 6A), and input/output ratios were comparable across conditions (Supplementary Fig. 6B).

After observing beige fat-dependent increases in microglial IL4Ra expression in early obesity (Fig. 1g), and beige adipose-dependent rescue of LTP following SAT transplantation (Fig. 3h), we hypothesized that beige adipocytes might rescue LTP via the central actions of IL4. For insight into this possibility, LTP recordings were performed after preincubation with an IL4 neutralizing antibody (IL4 nAb; 5.0 μg/mL). These experiments revealed that SAT transplantation normalizes LTP in an IL4-dependent manner (Fig. 3i, j). In ACSF, slice preparations from $TRANS_{nTg}$ mice exhibited LTP that was indistinguishable from LFD/SHAM, but neutralization of IL4 unmasked deficits that were comparable to HFD/SHAM (Fig. 3j). Application of IL4 nAb eliminated synaptic rescue following SAT transplantation without inducing deficits in LFD/SHAM slices (Fig. 3j), suggesting that IL4 might be involved in communication between beige adipocytes and hippocampal neurons.

**Beige adipocytes in subcutaneous fat alternatively activate microglia in obese mice**. To determine whether increased SAT mass alters microglial activation in obesity and examine the potential role of beige fat in these effects, we isolated FMCs from LFD/SHAM, HFD/SHAM, $TRANS_{nTg}$, and $TRANS_{Tg}$ mice by density gradient centrifugation, as described[15]. To focus on early-onset changes among resident microglia, mice were maintained on HFD or LFD for 2 wk before sham surgery or SAT transplantation, with cell isolation 2 wk after surgery, as shown (Fig. 4a). Cells were gated on size and granularity, followed by doublet discrimination and dead cell exclusion as shown (Fig. 1e). In our initial experiments, there was no evidence of macrophage infiltration in nTg mice maintained on HFD for 4 wk (Fig. 1f). However, we considered the possibility that surgery might unmask vulnerability to obesity-induced macrophage infiltration at the 4 wk time point. There was no effect of diet or SAT transplantation on the size of the $CD45^{HI}/Ly6C^{HI}$ population (% of CD11b, mean ± sem, n = 6/condition: LFD/SHAM = 8.5 ± 0.6; HFD/SHAM = 8.9 ± 0.7; $TRANS_{nTg}$ = 8.0 ± 1.1; $TRANS_{Tg}$ = 7.9 ± 0.8).

After observing no apparent shift in cellular phenotype with SAT transplantation, resident microglia were gated as $CD11b^+$/$CD45^{LO}/Ly6C^{LO}$ events for analysis of classical (M1) and alternative (M2) activation markers (Fig. 4b). Analysis of the M1 markers TLR4 and MHCII revealed that pro-inflammatory polarization persists in FMCs from SAT transplant recipients (Fig. 4b, c). However, subsequent analysis of M2 cytokine receptors revealed co-induction of interleukin-4 receptor-alpha (IL4Ra) in microglia from transplant recipients (Fig. 4b, d). Upregulation of IL4Ra was observed in FMCs from $TRANS_{nTg}$ mice, but not in cells from $TRANS_{Tg}$ mice (Fig. 4d). These effects were selective for IL4Ra, as there was no effect of diet or surgery on microglial IL10Ra (Fig. 4d).

Intracellular arginase-1 (Arg1) is a reliable downstream readout for IL4Ra-mediated anti-inflammatory activation[26]. After observing the co-induction of IL4Ra and the classical activation markers MHCII and TLR4, we measured intracellular Arg1 in FMCs as a downstream marker of M2 polarization. SAT transplantation significantly increased intracellular Arg1 fluorescence, and this

effect was dependent on beige adipocytes in the donor (Fig. 4e). To specifically look at hippocampal microglia, we visualized Arg1 using immunofluorescence. Analysis of Arg1 expression in $IBA1^+$ cells revealed significantly more $Arg1^+$ microglia in $TRANS_{nTg}$, relative to HFD/SHAM and LFD/SHAM mice (Fig. 4f). In our first set of experiments, analysis of microglial process architecture revealed no effect of diet in nTg mice after 4 wk HFD (Fig. 1h). However, the underlying mechanisms for morphological responses to HFD over time remain poorly understood. To gain insight into heterogeneous morphological responses, we analyzed microglial morphology separately for the $Arg1^-$ and $Arg1^+$ populations. Consistent with the anti-inflammatory M2 state, there was no effect of diet or surgery on the ramification of $Arg1^+$ microglia (Fig. 4g). By contrast, the subpopulation of $Arg1^-$ microglia exhibited reduced anatomical complexity in HFD/SHAM mice, relative to LFD/SHAM (Fig. 4h). Anatomical simplification of $Arg1^-$ microglia was eliminated by SAT transplantation, but only when transplants were collected from an nTg donor (Fig. 4h). These results indicate that beige adipocytes potentiate M2 polarization, and that the subset of alternatively activated microglia maintains a highly ramified morphology. Taken together, these results strongly implicate SAT beige adipocytes as an anti-inflammatory stimulus, with a potential role in the resolution of obesity-associated neuroinflammation and cognitive decline.

**Beige adipocytes promote IL4 induction in CNS T cells**. After observing the upregulation of microglial IL4Ra in early obesity and a requirement for IL4 in synaptic rescue after SAT transplantation, we investigated potential sources of IL4 in the CNS and periphery. Mice were maintained on HFD or LFD for 4 wk, with SAT transplantation from nTg or $Adiponectin^{cre}/Prdm16^{fl/fl}$ (Tg) donors during week 2, as shown (Fig. 4a). Multiplex analysis of circulating cytokines revealed increases in IL1b and TNFa with early obesity (Fig. 5a). Increases in circulating pro-inflammatory cytokines occurred irrespective of surgery or donor genotype (Fig. 5a). IL4 is produced at very low levels in unstimulated animals, and its rapid turnover limits detectability by standard immunoassays[27]. We, therefore, used an in vivo capture assay in which biotinylated antibodies were administered intraperitoneally for analysis of circulating IL4, or into the lateral cerebral ventricles for measures in CSF (Fig. 5b, c). Analysis of CSF collected from the cisterna magna 24 h later revealed increases in $TRANS_{nTg}$, but not in $TRANS_{Tg}$ mice (Fig. 5b). Increases in CSF IL4 were independent of circulating IL4 concentrations, which did not differ between diet and surgical conditions (Fig. 5c).

T cells in the meninges and choroid plexus are the primary source of IL4 under intact conditions[13]. To visualize meningeal T cells, mice were transcardially perfused after CSF collection and decalcified skulls were sectioned for immunofluorescence. $CD3e^+$ T cells were quantified along alpha laminin-labeled basement membranes in the choroid plexus and meninges. These experiments revealed the accumulation of T cells in both compartments with dietary obesity (Fig. 5d, g). While increases in T-cell number were unaffected by SAT transplantation, streptavidin amplification of the biotinylated IL4 capture antibody revealed significant increases in $IL4^+$ T cells in the choroid plexus and meninges (Fig. 5e, h). Induction of IL4 in meningeal T cells was observed in $TRANS_{nTg}$ mice, but not in $TRANS_{Tg}$ mice (Fig. 5d–h), suggestive of responses to beige adipose-derived signals. We cannot exclude the possibility that other lymphocytes in addition to T cells might be generating IL4, but it should be noted that $CD3e^-/IL4^+$ cells were rare (% of $IL4^+$, mean ± sem from n = 5–6/condition: LFD/SHAM = 0 ± 0; HFD/SHAM = 1.5 ± 1.5; $TRANS_{nTg}$ = 1.8 ± 0.9; $TRANS_{Tg}$ = 3.2 ± 2.1). $CD3e^+/IL4^+$ and $CD3e^-/IL4^+$ cells were localized to laminin-positive basement

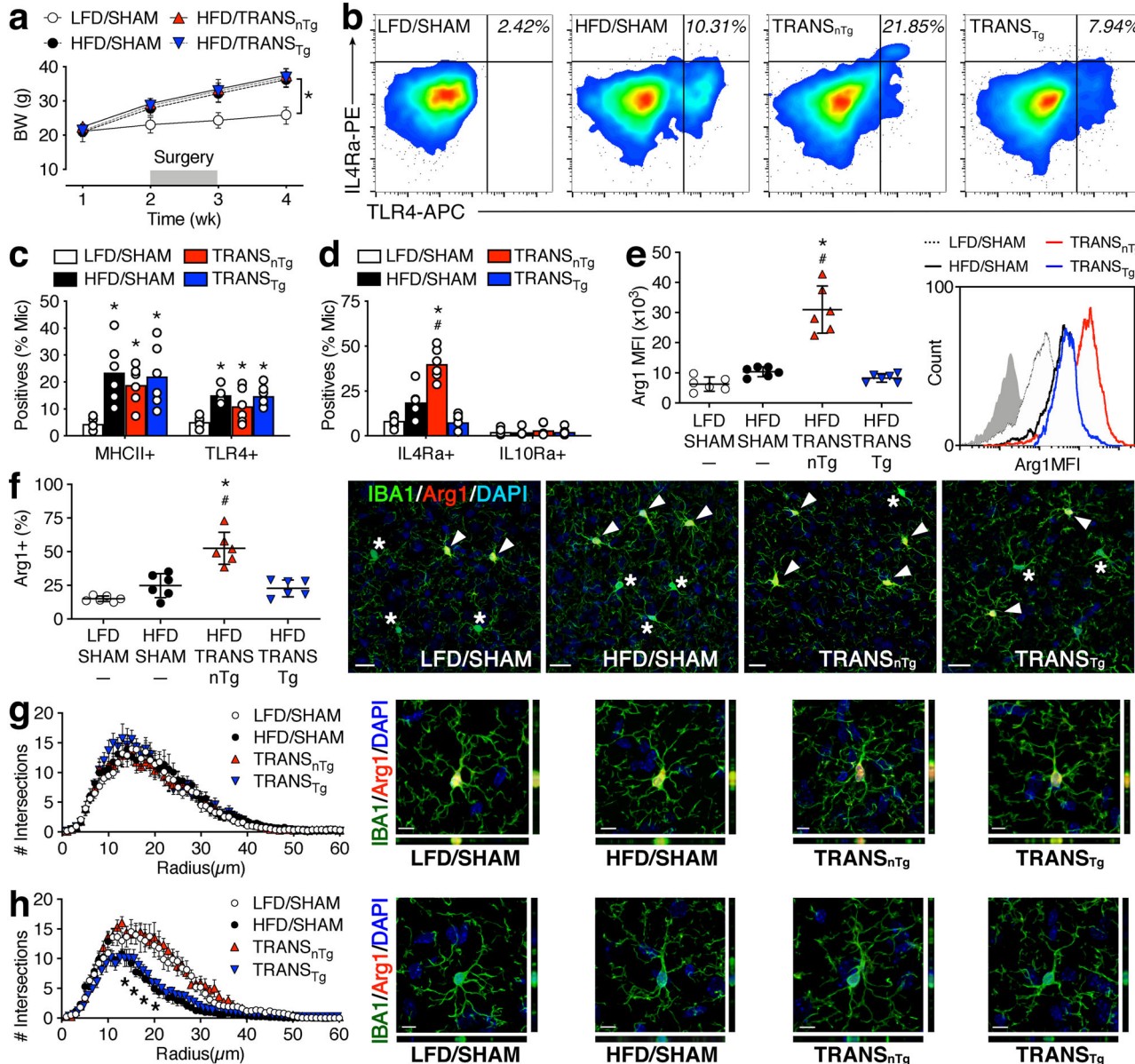

**Fig. 4 Beige adipocytes alternatively activate resident microglia in early obesity. a** Weight gain in Wt mice consuming a low-fat or high-fat diet (LFD, HFD) for 1 month, with a sham operation (SHAM) or transplantation of subcutaneous fat from a nontransgenic (TRANS$_{nTg}$) or Adiponectin$^{cre}$/PRDM16$^{fl/fl}$ transgenic donor (TRANS$_{Tg}$) during week 2. Symbols show group mean ± sem ($n = 12$ mice/condition). **b** Co-induction of classical (TLR4) and alternative activation-associated markers (IL4Ra) in CD11b$^+$/CD45$^{LO}$/Ly6C$^{LO}$/CD169$^-$/TMEM119$^+$ microglia from mice in the indicated conditions. Numbers in the upper-right quadrant represent TLR4$^+$/IL4Ra$^+$ events (% microglia). **c** Quantification of microglial MHCII and TLR4 revealed comparable pro-inflammatory polarization across all surgical conditions with dietary obesity. Symbols represent individual mice and bar height shows mean ± sem ($n = 6$ mice/condition; applies to **c, d**). **d** Beige adipose-dependent increases in microglial IL4Ra after SAT transplantation in HFD mice. **e** Left graph shows flow cytometric quantification of intracellular Arginase-1 (Arg1) mean fluorescence intensities (MFI) in gated microglia. Symbols represent individual mice and lines show mean ± sem ($n = 6$ mice/condition). Histogram (right) shows Arg1 MFI in gated microglia from representative samples in each condition (isotype shown in gray). **f** Quantification of Arg1 immunoreactivity in IBA1$^+$ microglia on hippocampal sections. Symbols represent individual mice and lines show mean ± sem ($n = 6$ mice/condition). Micrographs to the right of panels **f–h** depict representative labeling in the dentate molecular layer for the indicated groups. **g** Sholl analysis of Arg1$^+$ microglia. Symbols represent mean ± sem from $n = 6$ mice/condition (5 Arg1$^+$ cells per mouse). **h** Sholl analysis of Arg1$^-$ microglia reveals selective reductions in anatomical complexity that are alleviated by SAT transplantation from an nTg donor. Symbols represent mean ± sem from $n = 6$ mice/condition (5 Arg1$^-$ cells per mouse). Scale bar = 20 μm for micrographs in **f**; scale bar = 10 μm for **g, h**. For all graphs, white circles represent LFD/SHAM; black circles, HFD/SHAM; red triangles, TRANS$_{nTg}$; inverted blue triangles, TRANS$_{Tg}$. For **a**, repeated-measures ANOVA; for **c–h**, one-way ANOVA. *$p < 0.05$ relative to nTg/LFD; #$p < 0.05$ relative to nTg/HFD, determined by Tukey's multiple comparison test or Dunnett's T3 for heterogeneity of variance. For data, statistics, and exact $p$-values, see Source Data File 4.

membranes in the leptomeninges and choroid plexus, and were not observed in the CNS proper.

Mice used in the immunofluorescence experiments received intraventricular infusions of biotinylated antibodies against IL4.

Of necessity, this approach involves some limited damage to the meninges during the stereotaxic injection. We, therefore, carried out flow cytometry experiments to measure IL4 in T cells from a non-overlapping cohort of LFD/SHAM, HFD/SHAM, TRANS$_{nTg}$, and

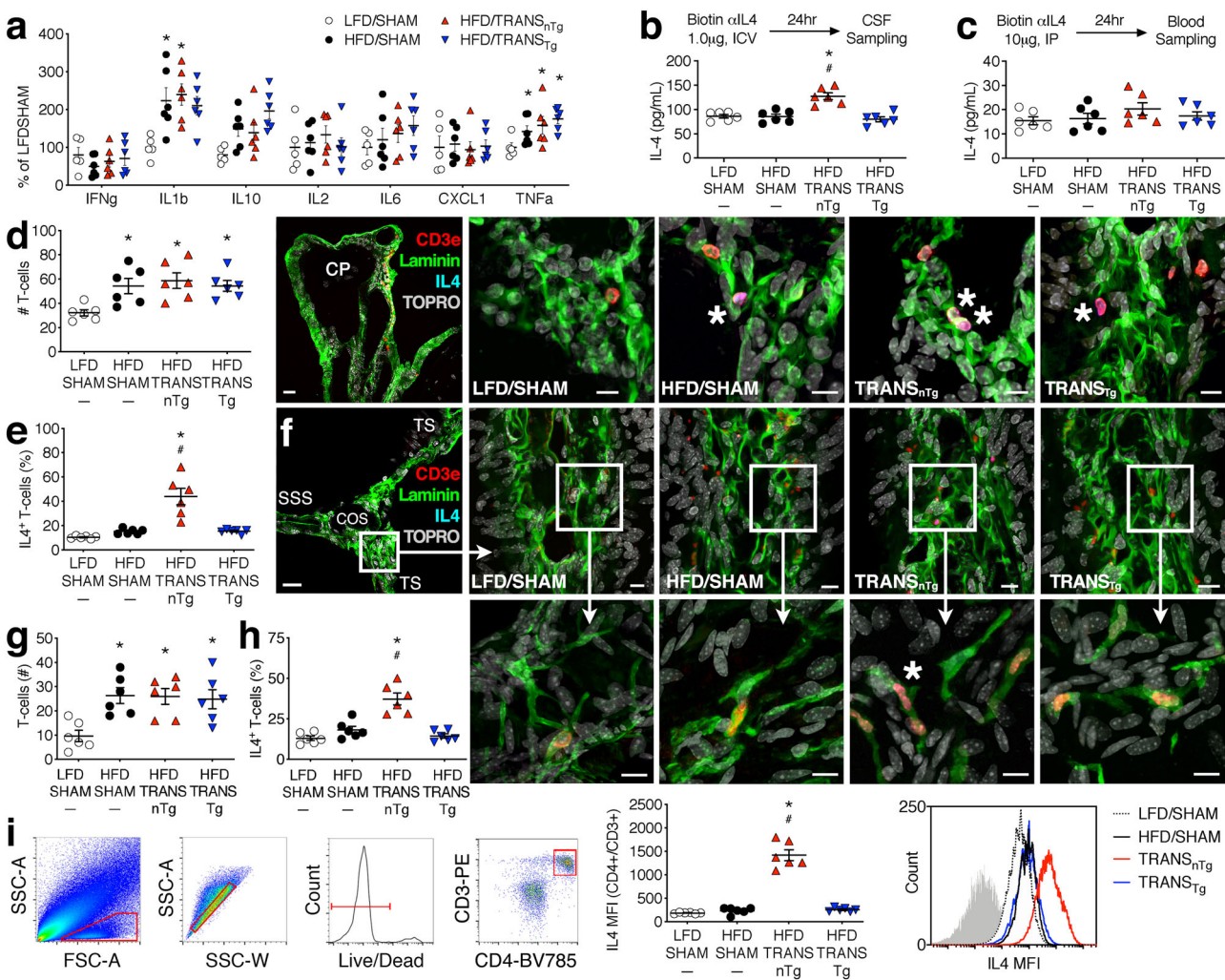

**Fig. 5 Surgical increases in subcutaneous fat increase IL4 in CNS T cells via beige adipose-dependent mechanisms. a** Multiplex analysis of serum cytokines after 4 wk on a low-fat or high-fat diet (LFD, HFD), with the sham operation (SHAM) or SAT transplantation from a nontransgenic (TRANS_nTg) or transgenic donor (TRANS_Tg) during week 2 (LFD/SHAM, $n = 5$; all other groups, $n = 6$). Symbols represent individual mice and lines show mean ± sem (applies to **a**, **i**). **b** Top panel shows schematic of in vivo antibody capture assay for detection of interleukin-4 (IL4) in cerebrospinal fluid (CSF). Graph (bottom) shows beige fat-dependent increases in CSF IL4 ($n = 6$ mice/condition). **c** Top panel shows in vivo capture approach for analysis of circulating IL4. Graph (bottom) shows quantification of serum IL4 by ELISA ($n = 6$ mice/condition). **d** Graph shows an increased T-cell number in the choroid plexus with dietary obesity ($n = 6$ mice/condition; applies to **d–h**). Micrographs (right) show representative labeling for laminin, CD3e, and biotinylated IL4 capture antibody in the choroid plexus (CP) following ICV delivery. Scale bar for anatomical reference micrograph (left) = 20 μm; for group comparison micrographs, scale bar = 10 μm. **e** Increased numbers of IL4+ T cells in the choroid plexus of TRANS_nTg mice. **f** Meningeal CD3e+ cells were quantified along laminin-labeled basement membranes in the transverse sinus (TS) adjacent to the confluence of sinuses (COS) and posterior to the superior sagittal sinus (SSS). Scale bar for anatomical reference (left) = 20 μm; for group comparison micrographs, scale bar = 10 μm. **g** Meningeal T-cell number increases with obesity, irrespective of SAT transplantation. **h** Increased uptake of IL4 capture antibody in meningeal CD3e+ T cells after SAT transplantation from a nontransgenic (TRANS_nTg) but not an Adiponectin^cre/PRDM16^fl/fl transgenic donor (TRANS_Tg). Micrographs (right) show immunofluorescence detection of biotinylated anti-IL4 in CD3e+ lymphocytes. **i** Gating strategy for analysis of endogenous IL4 expression in CNS T cells (CD4+/CD3e+). Graph (middle) shows increased intracellular IL4 fluorescence in T cells from TRANS_nTg mice, relative to all other groups ($n = 6$ mice/condition). Histograms (right) show IL4 mean fluorescence intensities (MFI) for representative samples in each condition (isotype control shown in gray). For all graphs, white circles represent LFD/SHAM; black circles, HFD/SHAM; red triangles, TRANS_nTg; inverted blue triangles, TRANS_Tg. For **a–i**, one-way ANOVA; *$p < 0.05$ relative to nTg/LFD; #$p < 0.05$ relative to nTg/HFD, determined by Tukey's multiple comparison test or Dunnett's T3 for heterogeneity of variance. For data, statistics, and exact $p$-values, see Source Data File 5.

TRANS_Tg mice that did not receive stereotaxic injections (Fig. 5i). Analysis of CD3e+/CD4+ cells revealed significantly brighter intracellular fluorescence in TRANS_nTg samples relative to all other groups (Fig. 5i). While antibodies against other immunophenotypic markers were not included in this panel, intracellular IL4 fluorescence was considerably lower among CD3e−/CD4− live cells, relative to CD3e+/CD4+ cells, and did not differ between experimental

conditions (IL4 fluorescence intensity [AU], mean ± sem from $n = 6$/condition: LFD/SHAM = 158 ± 11.8; HFD/SHAM = 168 ± 9.0; TRANS_nTg = 146 ± 10.5; TRANS_Tg = 152 ± 13.8). Together with the immunofluorescence results, the flow cytometry data implicate CNS T cells as a source for increases in CSF IL4 concentrations.

Under normal conditions, parenchymal IL4 concentrations are low or undetectable[28,29]. However, because the local synthesis of

IL4 in the brain parenchyma has been reported in models of encephalitis and acute ischemic injury[19,30], we measured *Il4* gene expression in primary astrocytes, microglia, and brain vascular endothelial cells (BVECs), as described[15]. Cells were plated and maintained overnight in the presence or absence of the allergenic agent ovalbumin (OVA, 100 μg/mL). As reported previously for freshly isolated glia and vascular cells from the intact brain[28,29], we were unable to reliably amplify *Il4* mRNA from unstimulated cells in these experiments (Supplementary Fig. 7A). Overnight exposure to OVA-induced robust IL4 mRNA expression in all cell types, but there was no effect of diet or surgery (Supplementary Fig. 7B). Stimulation with OVA induces expression of both anti- and pro-inflammatory cytokines, including interleukin-1beta (*Il1b*)[31]. Ovalbumin-stimulated induction of *Il1b* was increased in microglia from mice with dietary obesity, relative to lean mice, but increases were comparable in cells from sham-operated mice and SAT transplant recipients (Supplementary Fig. 7C). There were no differences in cell viability after OVA, as determined by formazan cleavage assay (Supplementary Fig. 7D).

**Recipient T cells are required for the effects of SAT transplantation on cognition and microglial activation**. After observing increases in CSF IL4 and accumulation of CNS T cells in SAT transplant recipients, we considered the following scenarios: first, lymphocytes from the SAT donor could be trafficking into the CNS; and second, lymphocytes from the recipient could interact with beige adipocytes in a manner that enhances cognition and reduces neuroinflammation. To evaluate the first scenario, we performed SAT transplantation using donor mice with ubiquitous inducible expression of tdTomato (Fig. 6a). Reporter expression was driven by a hybrid CAGG promoter fused to creER (CAGG^creER)[32]. Consistent with published expression data from this model, widespread tdTomato fluorescence was evident 2 wk after tamoxifen administration (3 × 2.0 mg every 48 h, PO). To examine immune cell trafficking between fat and brain, mice were maintained on LFD or HFD for 10 wk before receiving SAT transplants from a lean CAGG^creER/tdTomato^fl/fl donor. Two weeks after SAT transplantation, blood, stroma–vascular fraction (SVF), and FMCs were isolated and stained for flow cytometry. Cells were gated on size and granularity, followed by doublet discrimination and dead cell exclusion. T cells were gated as CD45^−/CD3e^+/CD4^+ events and monocytes were gated as CD45^+/CD11b^+ as shown (Fig. 6b). tdTomato^+ events were quantified relative to total T cells or monocytes as a measure of lymphocyte trafficking. These experiments revealed no evidence of lymphocyte trafficking between transplanted SAT and the CNS (Fig. 6c). tdTomato^+ events comprised <0.5% of CNS T cells and <0.25% of CNS monocytes, and the proportion of tdTomato^+ cells in each gate was similar in LFD and HFD samples (Fig. 6c). Circulating tdTomato^+ lymphocytes were also scarce, and there was no effect of dietary obesity (Fig. 6c). Viable tdTomato^+ T cells and monocytes were present in SVF from the transplanted SAT (Fig. 6c), and infiltration of recipient (tdTomato^−) lymphocytes into the transplant was unaffected by diet 2 wk after surgery (Fig. 6c). Relative to the CNS, lymphocyte trafficking between resident adipose depots and transplanted SAT was more frequent (Fig. 6c). Transplants were implanted into the peritoneal cavity, and interstitial continuity with resident epididymal fat likely accounted for the accumulation of tdTomato^+ T cells and monocytes in visceral fat. Dietary obesity was associated with increased numbers of tdTomato^+ T cells in resident VAT (Fig. 6c). In resident SAT, tdTomato^+ lymphocytes were less frequent than in VAT, and the proportions of tdTomato^+ T cells and monocytes were unaffected by diet (Fig. 6c). Taken together,

these data suggest that trafficking of donor lymphocytes between transplanted SAT and the CNS is unlikely.

To determine whether recipient-derived lymphocytes might mediate communication between beige adipocytes in transplanted SAT and the CNS, we carried out additional experiments using scid mice on the C57Bl6 background (B6.scid). Homozygous B6.scid mutant mice were maintained on HFD for 10 wk before SAT transplantation or sham operation, with parallel groups of mice maintained on LFD before sham operation. Weight gain was increased with HFD consumption, but there was no effect of surgery (Fig. 6d). Analysis of circulating cytokines revealed increased serum IL1b with dietary obesity in B6.scid mice (Fig. 6e), consistent with the persistent pro-inflammatory effects of high-fat diet reported by other groups using this model[33].

Two weeks after surgery, mice were tested in the water maze before transcardial perfusion and immunofluorescence visualization of hippocampal microglia. Testing in the water maze revealed a requirement for recipient-derived lymphocytes in the effects of SAT transplantation on cognition (Fig. 6f). HFD/SHAM and HFD/TRANS scid mice had longer path lengths than LFD/SHAM scid mice during acquisition training (Fig. 6f; $F_{2,33} = 5.60$, $p < 0.01$), and both groups of HFD mice spent less time swimming in the platform quadrant during the probe trial (Fig. 6f; $F_{2,33} = 12.17$, $p < 0.01$). Quantification of microglial process architecture revealed a similar loss of responsiveness to SAT transplantation (Fig. 6g). Three months on HFD reduced microglial process length and complexity in B6.scid mice ($F_{2,12} = 4.97$, $p < 0.05$), and reductions were comparable in HFD/SHAM and HFD/TRANS (Fig. 6g). Taken together, these data indicate that recipient-derived lymphocytes rescue cognition and restore microglial quiescence following SAT transplantation.

## Discussion

This study has identified an immunoregulatory relationship between beige adipocytes in subcutaneous fat, peripheral lymphocytes, and parenchymal cells in the hippocampus. Mice lacking beige adipocytes exhibited stronger pro-inflammatory responses to a high-fat diet in the brain and periphery, and were more susceptible to cognitive deficits and hippocampal synaptic dysfunction. Subcutaneous fat transplantation restored hippocampus-dependent memory in Wt recipients with dietary obesity and induced a unique microglial phenotype characterized by co-induction of pro- and anti-inflammatory markers. These effects were dependent on the presence of beige adipocytes in the donor, and electrophysiological recordings implicated IL4 signaling downstream of beige fat-mediated reinstatement of LTP. Beige adipocytes were required for induction of IL4 among lymphocytes in the meninges and choroid plexus, and recipient-derived lymphocytes were required for the cognitive effects of SAT transplantation. Lymphocyte interactions with beige adipocytes may therefore underlie the neuroprotective effects of subcutaneous adiposity, with a potential role for microglia as sensors and transducers in the CNS.

Visceral adiposity is a major risk factor for obesity-induced metabolic dysfunction, and mounting evidence supports similar associations with cognition[4–6,15]. Subcutaneous adiposity is considered metabolically benign[7], but relative to visceral fat, less is known regarding relationships with neuroinflammation and memory. In the current report, we observed that SAT becomes more "visceral-like" in the absence of beige fat, based on induction of pro-inflammatory genes in SAT from Adiponectin^cre/Prdm16^fl/fl mice with dietary obesity. However, changes in peripheral inflammation were not required for cognitive rescue, as SAT transplantation normalized hippocampal function without attenuating obesity-induced elevations in circulating

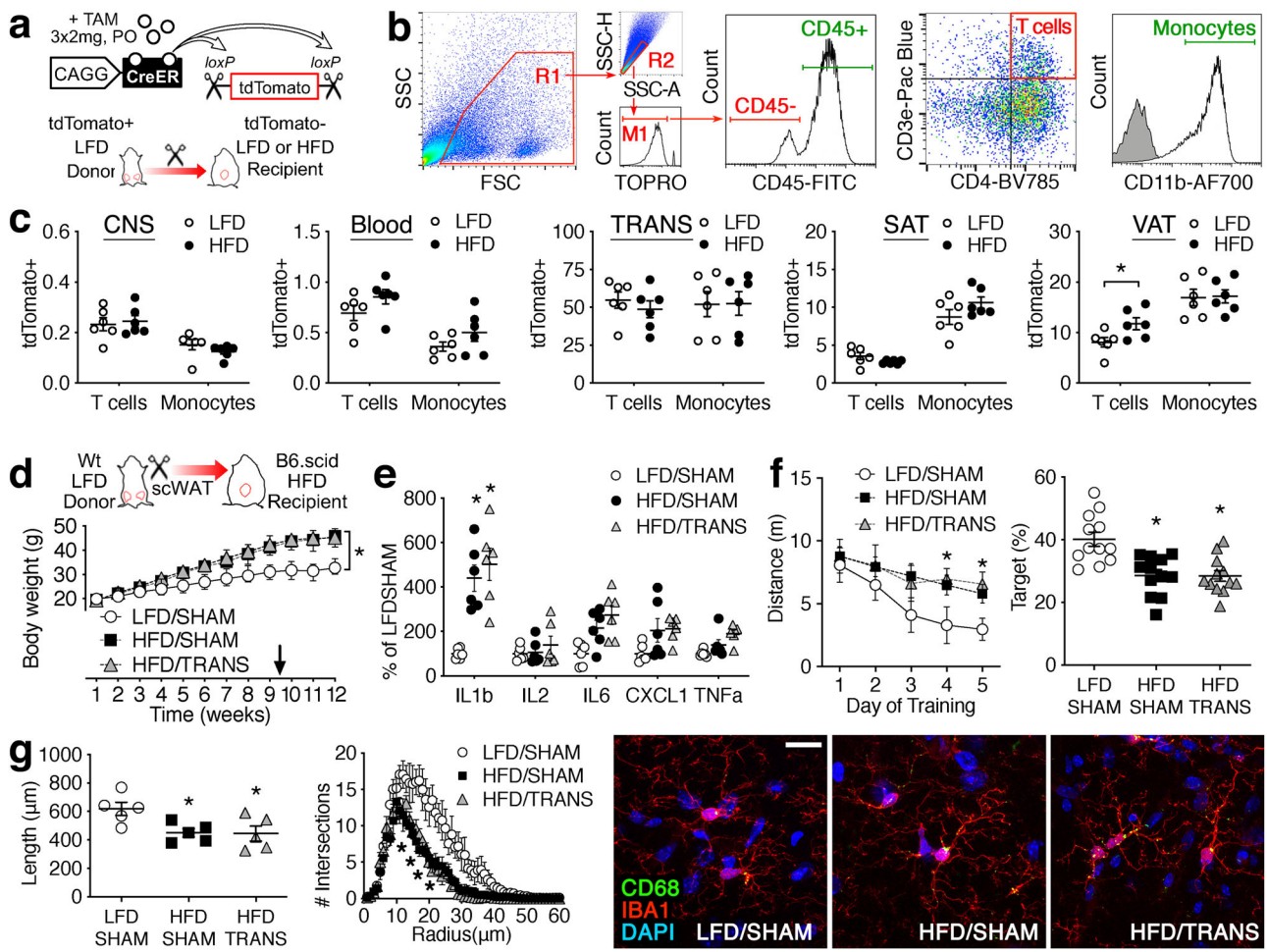

**Fig. 6 Recipient lymphocytes are required for the effects of SAT transplantation on cognition. a** Top panel shows a schematic of transgenic mice with inducible ubiquitous expression of tdTomato. Reporter mice were used as subcutaneous fat donors for wildtype (Wt) recipients on a low-fat or high-fat diet (LFD, HFD). **b** Gating strategy for analysis of lymphocyte trafficking in forebrain mononuclear cells and stromal–vascular fraction (SVF) from resident and transplanted adipose tissue. **c** tdTomato$^+$ T cells (CD45$^-$/CD3e$^+$/CD4$^+$) and monocytes (CD45$^+$/CD11b$^+$) were rare in the central nervous system (CNS) of transplant recipients, and their frequency was unaffected by recipient diet. Donor-derived T cells and monocytes were also rarely detected in lysed whole blood, and there was no effect on the recipient diet. Numerous viable tdTomato$^+$ T cells and monocytes were detected in SVF from subcutaneous fat transplants (TRANS). A smaller population of tdTomato$^+$ cells was detected in resident subcutaneous adipose tissue (SAT), with no effect of diet. In epididymal visceral adipose tissue (VAT), dietary obesity was associated with the accumulation of tdTomato$^+$ T cells. For each graph, symbols represent individual LFD recipients (white circles) or individual HFD recipients (black circles). Lines show mean ± sem from ($n = 6$) mice/ condition. **d** Top panel shows the experimental design for SAT transplantation experiments in B6.scid mutant mice. Bottom graph shows weight gain in B6. scid mutant mice on LFD or HFD, with the sham operation or SAT transplantation from a Wt donor during week 9 of the experiment (arrow). Symbols represent mean ± sem ($n = 12$ mice/condition). For **d**–**g**, white circles represent LFD/SHAM; black circles, HFD/SHAM; gray triangles, HFD/TRANS. **e** Multiplex analysis of serum cytokines in B6.scid mice. Symbols represent individual mice and lines show mean ± sem ($n = 6$ mice/condition). **f** Persistent deficits in water maze acquisition (left) and probe trial performance (right) in B6.scid mice after SAT transplantation. For acquisition (left), symbols represent group mean ± sem; for probe trial (right), symbols represent individual mice and lines show mean ± sem ($n = 12$ mice/condition). **g** Comparable reductions in microglial process length (left) and complexity (right) in HFD/SHAM and HFD/TRANS B6.scid mice. For process length (left), symbols represent individual mice and lines show mean ± sem ($n = 5$ mice/condition). For complexity (right), symbols indicate group mean ± sem from ($n = 5$) mice/condition. For both graphs, (5) cells were analyzed from each mouse. Micrographs (far right) depict representative images of IBA1$^+$ microglia in the hippocampal dentate molecular layer. Scale bar (LFD/SHAM) = 20 microns and applies to all. **c** *$p < 0.05$, bidirectional $t$-test. For **d**, **f** [left], repeated-measures ANOVA; for **e**, **f** [right], **g** [left], one-way ANOVA; *$p < 0.05$ relative to LFD/SHAM determined by Tukey's multiple comparison test or Dunnett's T3 for heterogeneity of variance. For data, statistics, and exact $p$-values, see Source Data File 6.

pro-inflammatory cytokines. Co-induction of pro- and anti-inflammatory markers by resident microglia emerged as a signature of cognitive rescue in obese mice, and the dissociation between microglial activation and circulating cytokines suggested an alternative site for stimulation of this unique phenotype.

Cold-stimulating beiging in SAT is dependent on the anti-inflammatory cytokine IL4[9], and exercise-induced myokine release also promotes beiging via IL4-dependent mechansims[8].

In the current report, cognitive rescue in SAT transplant recipients was dependent upon the presence of beige adipocytes in the donor, with correlated changes in CSF (but not circulating) IL4. Saturable transporters at the blood–brain and blood-CSF barriers have been identified and characterized for other members of the interleukin family, but transport mechanisms governing the entry and exit of IL4 into the brain have yet to be identified. Induction of IL4 among cells of the brain parenchyma has been

reported after ischemia and in experimental autoimmune encephalitis[19,30], but the parenchymal expression and brain-to-CSF efflux of IL4 was unlikely in the current experiments based on the absence of detectable IL4 gene expression under unstimulated conditions in primary microglia and astrocytes. Given the lack of IL4 mRNA expression among brain vascular endothelial cells, and the absence of changes in circulating IL4, blood-to-brain IL4 signaling was unlikely in these studies. The absence of IL4 expression in parenchymal and vascular cells, together with the donor genotype-dependent accumulation of IL4 in CSF, suggested that IL4 is being generated by cells with peripheral origins, or by cells of the leptomeninges and choroid plexus epithelium in response to a beige fat-dependent peripheral signal.

T-cell-derived IL4 in the meninges and choroid plexus supports learning and memory, based on deficits in water maze performance following antibody-mediated depletion of CSF T cells and bone marrow chimeras[34]. In the current report, beige adipocytes in transplanted SAT were required for IL4 induction among T cells in the meninges and choroid plexus, even after accounting for increases in T-cell number with dietary obesity. These findings implicate CSF T cells as a source of IL4, but T cells originate in the thymus, where they differentiate and mature from hematopoietic precursors before emigrating via the vasculature. Although routes of entry and egress into the CNS remain incompletely understood, T cells are thought to enter via the pial vasculature and exit via lymphatic drainage into the deep cervical lymph nodes[35,36]. Adding to the complexity, leptomeninges are not necessarily the "first stop" for newly mature T cells and other lymphocyte populations, as trafficking between the gut and meninges determines neurological outcome in models of ischemia and microbial infection[37,38]. In the current studies, we determined that lymphocyte trafficking between the transplanted SAT and the CNS was unlikely, as transplants from donor mice with whole-body expression of tdTomato did not yield considerable numbers of tdTomato[+] T cells or monocytes in the CNS. However, findings from SAT transplantation experiments in *scid* mutant mice with dietary obesity indicate that recipient-derived lymphocytes were required for improvements in memory and reductions in microglial activation. Taken together, these patterns suggest that host lymphocytes interact with beige adipocytes in transplanted SAT, and that these interactions regulate cognition and neuroinflammation.

The IL4 receptor-alpha chain (IL4Ra) is expressed by neurons, microglia, and astrocytes[26,39,40]. However, downstream signaling differs between microglia and other CNS cell types. In microglia, IL4Ra activation results in dimerization with the common gamma-chain of another ILR, forming the type 1 signaling complex[10]. In neurons and astrocytes, activation of IL4Ra leads to dimerization with interleukin-13 receptor alpha1 (IL13Ra1), forming the type 2 signaling complex[40]. The cellular consequences of both receptor complexes are anti-inflammatory, but the signaling cascades recruited by type 1 or type 2 signaling are distinct. The type 1 receptor complex recruits insulin receptor substrate (IRS) 1/2, while both receptor complexes activate JAK1/STAT6[10,11]. Signaling downstream of IRS1/2 activates kinases that enhance LTP, including Pi3K/AKT and ERK1/2[41]. Activation of JAK1/STAT6 has not been directly linked with synaptic plasticity, but other members of the JAK/STAT signaling pathway are selectively required for hippocampal long-term depression, and do not alter LTP[42]. In this report, we observed rapid reinstatement of LTP in slices treated with minocycline, and rapid elimination of protection in SAT transplant recipients following incubation with an IL4 scavenging antibody. While temporal kinetics do not demonstrate cell-type specificity, microglia represent a more likely cellular substrate than astrocytes given the different receptor complexes recruited by IL4 in the two cell types

and the distinct effects of downstream signaling cascades on LTP and LTD.

The overarching trend emerging from studies of adipose tissue distribution and cognition appears to recapitulate patterns from studies of insulin resistance and dysfunctional lipid metabolism, with a protective role for subcutaneous fat and deleterious effects of visceral adiposity. However, the relative inaccessibility of the CNS for longitudinal studies is a barrier to identifying, understanding, and ultimately treating the underlying cellular interactions that drive vulnerability to cognitive decline and dementia in obesity. For example, microglial interactions with cortical synapses and amyloid plaques have been visualized using longitudinal 2-photon imaging in lean mice[43,44], but the pro-inflammatory systemic effects of obesity are likely to interact with established methods for in vivo imaging[45]. Even the thin-skull preparation, which elicits less inflammation than cranial window imaging in lean adult mice[46,47], is likely to be complicated by the disordered bone formation and turnover in the calvaria with chronic obesity[48]. In terms of translatability, MRI and PET imaging techniques would enable direct comparison of data from humans and rodents, but the resolution of MRI and PET imaging and the small size of the mouse brain limits the utility of this widely-used research model. As transgenic rats become more widely available, longitudinal mechanistic obesity studies will likely provide new insights into the pathogenesis of obesity-induced cognitive dysfunction. Understanding the temporal etiology of circuit dysfunction with obesity could potentially uncover critical windows for noninvasive manipulation of neural activity, with the goal of attenuating risk of age-related cognitive decline and dementia in obese individuals.

## Methods

**Animals and diets**. Animals were housed in a ventilator rack on Alpha-dry bedding in a specific pathogen-free facility with food and water available ad libitum. The colony room was maintained on a 12 h light:dark cycle (lights-on at 0600 h). Colony temperature was set at 22 °C and recorded temperatures ranged from 21 to 24 °C over the course of the experiments. Breeding pairs of Adiponectin[cre]/Prdm16[fl/fl] mice[16] were imported from the laboratory of Bruce Spiegelman and bred in-house for these studies. All other mouse lines were purchased from Jackson Labs (reporter line Ai14, strain #007914; CAGG[creER] mice, strain #004682; B6.scid, strain #001913). Male transgenic mice and nTg littermates were maintained on LFD (Research Diets D12450) or HFD (Research Diets D12492) beginning at 8 wk old. Donor mice (6–8 wk old) for SAT transplantation experiments were maintained on standard chow (Teklad). For transgene induction, mice received 2.0 mg tamoxifen by oral gavage every 48 h for 6 d. Body weights were determined weekly for all experiments. All experimental procedures followed NIH guidelines and were approved by the Institutional Animal Care and Use Committee at Augusta University.

**Adipose transplantation surgery, adipose histology, and adipose gene expression**. For SAT transplantation, recipients and donors were maintained under Isoflurane anesthetic as described in Supplementary Methods and as reported[49]. Adipose tissue histology and gene expression assays are described in Supplementary Methods and followed published protocols[15,49]. For a list of Taqman probes used in gene expression assays, see Supplementary Table 1. For a list of antibodies and concentrations used for immunohistochemistry (Supplementary Table 2).

**Behavioral tests**. Cognitive testing in the water maze, novel object preference, and Y-maze was carried out during the first half of the dark cycle (1800–2200 h), as summarized in Supplementary Methods and as reported[15]. For the Barnes maze, mice were tested on a white circular platform with 12 holes evenly spaced around the perimeter. Testing was carried out over 7 d as described in Supplementary Methods, and as shown (Supplementary Fig. 8).

**Electrophysiology**. Acute slices were prepared on a Vibratome and allowed to recover for at least 1 h in carboxygenated artificial cerebrospinal fluid (ACSF) before recording, as described in Supplementary Methods and as reported[15,49]. For some experiments, slices were pre-incubated with minocycline (20 µM) or anti-IL4 neutralizing antibody (IL4 nAb, 5 µg/mL; rat anti-mouse IL4, clone 11B11; Invitrogen, Carlsbad, CA) for 30 min with continuous application throughout the recording. For additional details, see Supplementary Methods.

**Glucose metabolism and serum cytokines**. Glucose tolerance testing was carried out as described in Supplementary Methods and as reported previously[49,50]. Multiplex analysis of cytokines was performed using the Mesoscale Discovery platform (V-PLEX Pro-inflammatory Panel 1, Meso Scale Diagnostics, Rockville, MD) according to the manufacturer's instructions. Serum samples were assayed in duplicate on a QuickPlex SQ120 reader at the Emory University multiplex immunoassay core facility. To quantify IL4 in serum and CSF, we used an in vivo capture assay modified from Finkelman and Morris[27] (details in Supplementary Methods).

**Immunofluorescence and imaging**. For immunofluorescence visualization of microglia, 40-micron coronal sections were cut as a 1:6 series using a freezing microtome (Leica), as reported in Supplementary Methods and as described[15]. Immunofluorescence and morphological reconstruction procedures are outlined in Supplementary Methods. Protocols for decalcification, sectioning, and immunofluorescence analysis of meningeal T cells are also in Supplementary Methods. For a list of antibodies and dilutions, see Supplementary Table 2.

**Cell isolation and flow cytometry**. Cell isolation from brain and SVF was performed as described in Supplementary Methods and as reported[15]. Flow cytometric data acquisition was carried out on a 5-laser BD LSRII in FACSDiva software version 8.1 (BD Bioscience), a Guava EasyCyte 5.0 with Incyte (version 2.6, Millipore), or on an Accuri C6 flow cytometer with Accuri C6 Plus software (version 3.4, BD Bioscience). For each combination of antibodies, compensation parameters were determined using unlabeled cells, single-labeled samples, and isotype controls. For a list of antibodies used in flow cytometry experiments, see Supplemental Table 3.

**Ex vivo stimulation and qPCR**. For ovalbumin stimulation, FMCs were plated ($10^5$ cells/well) and stimulated with 100 micrograms/mL ovalbumin (Sigma-Aldrich; see Supplementary Methods). Methods for RNA extraction, cDNA synthesis, and qPCR are summarized in Supplementary Methods and followed published protocols[15]. For ovalbumin stimulation, expression was determined by calculating ddCT, with the average dCT from LFD/SHAM cells as the reference group. For paraffin-embedded adipose tissues, RNA extraction was carried out as described[15], with modifications (Supplementary Methods). cDNA samples were amplified using Taqman probes, with the average dCT for nTg/LFD (Supplementary Figs. 1 and 3) or nTg donor samples (Supplementary Fig. 5) as the reference for calculating relative expression. See Supplementary Table 1 for a list of Taqman probes used in these experiments.

**Statistics**. Experimental data from Adiponectin$^{cre}$/Prdm16$^{fl/fl}$ mice and littermate controls maintained on HFD or LFD were analyzed using $2 \times 2$ ANOVA or $2 \times 2$ repeated-measures ANOVA with Tukey's post hoc. For the SAT transplant experiments, the effects of diet, surgery, and donor genotype were analyzed using one-way ANOVA with repeated measures where appropriate, followed by Tukey's post hoc. Heterogeneity of variance was addressed using the Geisser-Greenhouse correction or Welch's correction, as appropriate. Lymphocyte trafficking was compared across LFD and HFD recipients using a bidirectional $t$-test for normally distributed data or a Mann–Whitney $U$-test for heterogeneity of variance. Statistical analyses were carried out in Graphpad Prism version 8.0 with statistical significance at $p < 0.05$.

**Reporting summary**. Further information on research design is available in the Nature Research Reporting Summary linked to this article.

## Data availability

Detailed step-by-step protocols are available on request from the corresponding author. Source data are provided with this paper.

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

## Acknowledgements
These studies were supported by grants from the National Institutes of Health to A.M.S. (K01DK100616, R03DK101817, R01DK110586). We are grateful to Dr. Fred Finkelman for advice on the IL4 capture assay, and to Dr. Paul Cohen for breeding pairs of Adiponectin^cre/Prdm16^fl/fl mice and comments on the manuscript.

## Author contributions
All authors were directly involved in data acquisition and analysis. D.-H.G., B.B., and A.M.S. designed experiments. D.-H.G. and A.M.S. prepared the manuscript and all authors provided comments on subsequent drafts.

## Competing interests
The authors declare no competing interests.
