## [Peer Review File · Nature Communications]

Reviewers' Comments:

Reviewer #1:

Remarks to the Author:

This paper claims that beige adipocytes are linked to hippocampal functioning through lymphocyte driven IL4 signaling. The claims appear to be novel and helping to build on existing work on the field in that it offers insight into the anti-inflammatory properties of the beige adipocytes in the subcutaneous fat pad whereas some of the earlier work of this lab focused on pro-inflammatory pathways from visceral fat pad. This paper will be of interest to others in the field as it broadens the picture on the varying roles of fat type, thus it will help influence how the field will think about different types of fat and where that fat resides moving forward.

The claims are convincing. However, there were a few things that could have been done to strengthen the paper further. Analysis of the adipocyte types within the transplants and within the subcutaneous fat pads of the non-transgenic animals would have provided a more solid basis that it was the beige adipocytes at work. The failure of the current paper to provide this analysis leaves the reader to rely on trust in the transgenic model and prior literature rather than being able to firmly connect the results presented to only the beige adipocytes instead of the entire subcutaneous fat pad. Further, the inclusion of this analysis would provide a way from the authors to compare between animals the behavioral and biological data based on the beige adipocyte percentage.

Also, it was exceedingly hard to decipher how many animals were used in each of the paper's experiments. While most of the figures had individual symbols denoting each animal, overlapping symbols and a lack of this information in any other form within the paper meant that it was up to the reader to try to decipher this information. Given that the trust in the appropriateness of the statistical tests and strength the data relies on the appropriateness of the group sizes, it would be useful for the authors to include the group numbers in a more direct forms, either in the paper body or in the supplemental materials.

The framing in the introduction does not provide enough of a solid ground for the reader to contextualize much of the results until they reach the discussion. I would suggest reworking the provided background information to provide more of the background information in the introduction with any repeated citations, if necessary, in the discussion to aid the reader. In this same frame, the paragraph on lines 467-492 could be eliminated to shorten the communication as it did not add to the findings nor help contextualize the findings.

On line 577, it is stated that the transplantation was carried out as previously reported in Guo et al 2020. However, that paper cites Erion et al 2014 rather than outlining the methodology in either the main text or the supplemental materials. Given this, it would be more appropriate to directly cite Erion et al 2014.

On line 608, there is a mention of extramaze cues used on days 3-6 of training on a Barnes maze. Details as to what these cues are, such as what it was, how far away it was, and where, are important given the work done by Prusky and others. See, for example, Prusky, West, and Douglas 2000 for the behavioral assessment of visual acuity in mice and rats.

On line 630, it is stated that glucose tolerance testing was carried out as previously reported, however the previous paper does not outline the methodology. Given that this paper properly outlines the methodology, this line needs to be deleted.

I do find that there is a lack of conclusiveness to the discussion in how this data ties into what is known about visceral fat and pro-inflammatory cytokine signaling. It may be that the authors were trying to avoid making bombastic claims, but it also left a hole in the conclusion to tie this paper to the existing literature in the field.

It is important to note that while this paper was obviously full of quite a bit of information, it was also very clearly written. It is accessible to scientists from a variety of levels and has a level of detail that would make the study replicable. The authors are careful not to oversell their claims while also being fair to the strength of their data. They would benefit from a bit more detail as noted above in my comments, but they do not have extraneous detail where it is not needed outside of the single paragraph noted.

The study holds sufficient promise that publication should be considered. Edits are required, but resubmission should be strongly encouraged.

Reviewer #2:

Remarks to the Author:

The manuscript specifically investigates the impact of subcutaneous fat on memory processes in different mice models. Based on evidence that subcutaneous fat is less pro-inflammatory than its visceral counterpart, the authors test the hypothesis that specific peripheral manipulations of beige adipocytes (i.e., through subcutaneous fat transplantation experiments) could have a beneficial effect on high-fat diet-induced cognitive decline through a microglial, cytokine signaling pathway. The results support the hypothesis presented in numerous ways, using appropriate statistical tests and several relevant animal models. This is a thorough and well-controlled study that investigates alternative hypotheses and potential confounds with great precision. It should be of significant interest to the readership of Nature Communication as it bridges successfully peripheral metabolic dysregulation with brain inflammation. The mechanisms outlined will help identify fundamental mechanisms aligning peripheral metabolic dysregulation with cognitive decline. The manuscript reads extremely well given the size of content provided and the points presented below are considered minor and to help clarify some interpretations.

1. The distinction between IL4KO and leukocyte depletion in younger animals suggesting that a functional immune system is critical to normal brain function and learning as well as hippocampal plasticity (lines 99-109), is difficult to reconcile with the discussion on lines 470-479 that with aging, upregulation of IL4 is implicated with cognitive decline. The authors take the stance that such controversies are due to the lack of standardized physiological exclusion criteria in aging studies, however, this is unlikely as the NIH has been requiring and implementing the use of more complete procedural descriptions of diet, environment, and the overall health status and body condition of each animal in studies of aging. Also, note that studies of aging in rats have been able to identify very different populations of animals: those that age with little if any cognitive decline, and those that age with large deficits. This is in line with clinical data and perhaps should be considered in the interpretation of the data.

2. While it is clear why the authors chose to emphasize aging-related evidence (i.e., diabetes, obesity, sedentary lifestyle, etc...) in the rationale for the study as well as in the Discussion section, the work presented was accomplished in young adult mice. This point does not detract from the clarity, novelty, and thoughtful characterizations presented here, nevertheless, a caveat should be included to mention that for accurate alignment with clinical manifestations of the disease processes engaged in T2DM, experiments in aged animals should be considered.

3. While the authors provide significant and substantive evidence that SAT transplantation attenuates brain neuroinflammation in obesity through an IL4 pathway, they do not provide evidence for the manner in which beige adipocytes from the transplant interact with the recipient's lymphocytes in the brain. The Discussion may need to be expanded to clarify this and offer the reader a tentative mechanism for how peripheral perturbations impact CNS function at the BBB.

4. Please clarify what is meant by "earlier onset" on line 150, as it is not clear whether this addresses animal ages or suggests time is a variable in these flow cytometry experiments. Also, please clarify line 209, as it not obvious from Figure 2E that the nTg/LFD show reductions in distance traveled, and no statistics are presented.

5. Given the size and complexity of the study, the authors should briefly describe the number of animals or cohorts that went into these experiments.

Olivier Thibault

Reviewer #3:

Remarks to the Author:

The manuscript by Guo et al. explores a possible protective role for inducible thermogenic fat (aka beige fat) in obesity-related cognitive impairment and neuroinflammation. Using an established mouse model with an ablation of beige fat function, the authors show that mutant mice have accelerated cognitive dysfunction and microglial activation in the setting of dietary obesity. The

authors then used a transplant model to show that subcutaneous fat (SAT) from wild-type donors can rescue these defects, while subcutaneous fat from mutant mice lacking functional beige fat cannot. Moreover, beige adipocytes are required for reinstating long-term potentiation, and the beneficial effects of SAT transplantation are eliminated upon treatment with an IL4 neutralizing antibody. The authors then show that beige fat induces IL4 production by CNS T cells. Using reporter mice, they show that these T cells do not seem to traffic from the periphery and that recipient T cells are required for the cognitive benefits associated with SAT transplantation.

This is a well written manuscript that represents an enormous body of work including mouse models, behavioral assays, cellular and molecular phenotyping, and electrophysiology. The findings highlight a novel role for beige fat in protecting against cognitive impairment related to obesity and provide a mechanism within the CNS that underlies these effects. As is true with any new finding, a large number of future experiments are now possible, but these are beyond the scope of this already very comprehensive study. Discussion of the following points would strengthen this already well done manuscript.

1) Throughout the text, the authors ascribe the effects to beige fat. While this may well be true, it seems there are at least two possibilities: (a) beige fat itself is neuroprotective or (b) in the absence of beige fat, subcutaneous fat acquires more visceral-like properties which would be deleterious for cognition. The ability of wild-type SAT transplants to correct the phenotype seems to argue more for possibility (a). However, wild-type SAT from room temperature housed mice has very few beige adipocytes, as these cells are typically induced in response to cold exposure or beta-adrenergic agonism. Have the authors considered testing the effects of SAT transplants from cold-exposed mice? It is possible that these transplants could have even stronger beneficial effects though it is unclear how long this phenotype might persist upon transplantation.

2) Aerobic exercise has also been shown to attenuate cognitive decline. Since exercise activates beige fat, the authors could consider in the future testing whether beige fat is required for the cognitive benefits linked to exercise. A similar type of transplant model has been employed to assess other benefits linked to beige fat (<https://pubmed.ncbi.nlm.nih.gov/26050668/>).

3) It would be nice if the authors speculated on possible mechanisms by which beige fat might mediate these changes in the brain. Do the authors believe there is a humoral factor that signals to the brain or a neuronal circuit?

4) The authors focus on microglia in this study, but have they considered a role for other cells that might respond to IL4, like astrocytes?

5) In the behavioral studies, the authors made efforts to account for factors like visual acuity and motor abilities, but not for anxiety levels. Is there any concern that there could be a difference in these groups?

We would like to thank the Reviewers for their time and energy providing feedback on the original submission. These comments have been addressed with additional data and complete reorganization of the Introduction and Discussion sections. Changes are addressed in a point-by-point manner below and are indicated with Track Changes in the revised submission.

Reviewer #1:

This paper claims that beige adipocytes are linked to hippocampal functioning through lymphocyte driven IL4 signaling. The claims appear to be novel and helping to build on existing work on the field in that it offers insight into the anti-inflammatory properties of the beige adipocytes in the subcutaneous fat pad whereas some of the earlier work of this lab focused on pro-inflammatory pathways from visceral fat pad. This paper will be of interest to others in the field as it broadens the picture on the varying roles of fat type, thus it will help influence how the field will think about different types of fat and where that fat resides moving forward. The claims are convincing. However, there were a few things that could have been done to strengthen the paper further.

R1.Q1. Analysis of the adipocyte types within the transplants and within the subcutaneous fat pads of the non-transgenic animals would have provided a more solid basis that it was the beige adipocytes at work. The failure of the current paper to provide this analysis leaves the reader to rely on trust in the transgenic model and prior literature rather than being able to firmly connect the results presented to only the beige adipocytes instead of the entire subcutaneous fat pad. Further, the inclusion of this analysis would provide a way from the authors to compare between animals the behavioral and biological data based on the beige adipocyte percentage.

- We agree that the presence of beige adipocytes in resident and transplanted SAT is a critical variable for these experiments. The revised submission includes qPCR analysis of genes associated with 'beiging' and anatomical/immunohistochemical markers of beiging in resident and transplanted SAT (Supplementary Figs.1 and 5).
- The Reviewer also emphasized the importance of internally replicating published data from other groups related to the adipose tissue phenotype in Adiponectin^{cre}/PRDM16^{fl/fl} mice. In response, we verified the integrity of interscapular brown adipose tissue (BAT) in Adiponectin^{cre}/PRDM16^{fl/fl} mice using HNE and UCP1 staining (Supplementary Fig.2A-B).

R1.Q2. Also, it was exceedingly hard to decipher how many animals were used in each of the paper's experiments. While most of the figures had individual symbols denoting each animal, overlapping symbols and a lack of this information in any other form within the paper meant that it was up to the reader to try to decipher this information. Given that the trust in the appropriateness of the statistical tests and strength the data relies on the appropriateness of the group sizes, it would be useful for the authors to include the group numbers in a more direct forms, either in the paper body or in the supplemental materials.

- We apologize for this omission. It was not our intent to obscure the n-sizes and degrees of freedom were reported for each statistic in the original submission. However it is true that some methods of correction for heterogeneity of variance (e.g. Greenhouse-Geisser) reduce degrees of freedom, limiting its informative value for determining group sizes. The revised submission includes the exact n-sizes for panels in each Figure in the Figure legend, and full datasets are provided as supplementary information.

R1.Q3. The framing in the introduction does not provide enough of a solid ground for the reader to contextualize much of the results until they reach the discussion. I would suggest reworking the provided background information to provide more of the background information in the introduction with any repeated citations, if necessary, in the discussion to aid the reader.

- The Introduction has been rewritten in order to present major themes and questions related to these experiments (pasted below). Where appropriate, minor themes are introduced with references under relevant sections of the Results.

" Unlike visceral fat, which contains a homogeneous population of white adipocytes, subcutaneous fat contains both white adipocytes and 'beige' adipocytes that expend energy in a manner analogous to brown fat (Cohen and Spiegelman, 2016). Beige adipocytes interact continuously with immune cells, and the acquisition of thermogenic features ('beiging') requires induction of the anti-inflammatory cytokines interleukin 4 (IL-4) by leukocytes in subcutaneous adipose tissue (SAT; Rao et al, 2014; Qiu et al, 2014). Given the importance of IL-4 signaling for allergic responses and autoimmunity, circulating concentrations of IL-4 are tightly regulated (Wills-Karp and Finkelman, 2008). However, tissue-resident immune cells are capable of eflux and migration to other organs, including the brain and leptomeninges (Friedl and Weigelin, 2008; Filiano et al, 2017). Immune cell trafficking and local signaling at the blood-brain and blood-cerebrospinal fluid interfaces therefore enables inter-organ crosstalk independently of circulating factors or direct access to the brain parenchyma (Filiano et al, 2017). Although peripheral macrophages gain access to the brain with chronic obesity, less is known about earlier neuroimmune interactions at the blood-brain and blood-cerebrospinal fluid interfaces (Buckman et al, 2014; Guo et al, 2020). Moreover, despite emerging roles in other chronic inflammatory diseases (Filiano et al, 2017), T-lymphocytes have received less attention with respect to their protective or pathogenic roles in the obese brain. We therefore investigated immunoregulatory interactions between beige adipocytes and cognition in a series of dietary obesity and SAT transplantation experiments. These studies indicate that beige adipocytes are indispensable for the neuroprotective and anti-inflammatory effects of subcutaneous fat, and implicate beige fat-stimulated IL-4 production by cerebrospinal fluid T-cells in communication between SAT and the CNS."

R1.Q4. In this same frame, the paragraph on lines 467-492 could be eliminated to shorten the communication as it did not add to the findings nor help contextualize the findings.

- The Discussion section has been completely revised and the section on aging has been deleted as suggested by the Reviewer. In place of this section, the Discussion includes paragraphs on peripheral interactions between subcutaneous and visceral fat (paragraph 2); relevant literature on beige fat and IL-4 in the brain and periphery (paragraph 3); and lymphocyte trafficking between brain and periphery (paragraph 4); and IL4 receptor-mediated signaling among cells of the brain parenchyma (paragraph 5).

R1.Q5. On line 577, it is stated that the transplantation was carried out as previously reported in Guo et al 2020. However, that paper cites Erion et al 2014 rather than outlining the methodology in either the main text or the supplemental materials. Given this, it would be more appropriate to directly cite Erion et al 2014.

- The reference has been updated as suggested. Both references (Guo et al, 2020 and Erion et al, 2014) outline procedures for VAT transplantation, and these published protocols were modified slightly for SAT transplantation in the current report. Modifications are described in detail in the Supplementary Methods section of the revised submission.

R1.Q6. On line 608, there is a mention of extramaze cues used on days 3-6 of training on a Barnes maze. Details as to what these cues are, such as what it was, how far away it was, and where, are important given the work done by Prusky and others. See, for example, Prusky, West, and Douglas 2000 for the behavioral assessment of visual acuity in mice and rats.

- The extramaze cues are described in detail in the Supplementary Methods section of the revised submission, and a schematic diagram of the apparatus was added (Supplementary Figure 8A-B).
- We agree that visual acuity is a critical variable for behavioral experiments. However, the mice used in the current report likely had normal vision based on the absence of group differences when swimming towards the visible platform in the water maze, and on the absence of group differences finding the escape hole based on a proximal visual cue in the Barnes maze. These data are reported in the text of the Results section.

R1.Q7. On line 630, it is stated that glucose tolerance testing was carried out as previously reported, however the previous paper does not outline the methodology. Given that this paper properly outlines the methodology, this line needs to be deleted.

- We apologize for the incorrect reference. That citation has been corrected (McGee-Lawrence et al, 2017 Endocrinology). In the interests of space, summarized methods for glucose tolerance testing were moved to the Supplementary Methods section in the revised submission.

R1.Q8. I do find that there is a lack of conclusiveness to the discussion in how this data ties into what is known about visceral fat and pro-inflammatory cytokine signaling. It may be that the authors were trying to avoid making bombastic claims, but it also left a hole in the conclusion to tie this paper to the existing literature in the field.

- In the revised submission, we discuss potential interactions between subcutaneous and visceral fat. However, we would like to respectfully emphasize that virtually all of previous work was carried out from the perspective of metabolic dysfunction. Thus far, most findings related to the regulation of metabolic homeostasis by visceral and subcutaneous fat have led to conceptually similar outcomes in the brain, but studies of cognition in preclinical models of obesity remain primarily correlative. That limitation on the field was our rationale for not addressing this issue in greater detail in the original submission.

R1.Q9. It is important to note that while this paper was obviously full of quite a bit of information, it was also very clearly written. It is accessible to scientists from a variety of levels and has a level of detail that would make the study replicable. The authors are careful not to oversell their claims while also being fair to the strength of their data. They would benefit from a bit more detail as noted above in my comments, but they do not have extraneous detail where it is not needed outside of the single paragraph noted. The study holds sufficient promise that publication should be considered. Edits are required, but resubmission should be strongly encouraged.

- We thank the Reviewer for their insightful and enthusiastic comments.

Reviewer #2:

The manuscript specifically investigates the impact of subcutaneous fat on memory processes in different mice models. Based on evidence that subcutaneous fat is less pro-inflammatory than its visceral counterpart, the authors test the hypothesis that specific peripheral manipulations of

beige adipocytes (i.e., through subcutaneous fat transplantation experiments) could have a beneficial effect on high-fat diet-induced cognitive decline through a microglial, cytokine signaling pathway. The results support the hypothesis presented in numerous ways, using appropriate statistical tests and several relevant animal models. This is a thorough and well-controlled study that investigates alternative hypotheses and potential confounds with great precision. It should be of significant interest to the readership of Nature Communication as it bridges successfully peripheral metabolic dysregulation with brain inflammation. The mechanisms outlined will help identify fundamental mechanisms aligning peripheral metabolic dysregulation with cognitive decline. The manuscript reads extremely well given the size of content provided and the points presented below are considered minor and to help clarify some interpretations.

- We thank the reviewer for their laudatory comments and for constructive feedback on this submission.

R2.Q1. The distinction between IL4KO and leukocyte depletion in younger animals suggesting that a functional immune system is critical to normal brain function and learning as well as hippocampal plasticity (lines 99-109), is difficult to reconcile with the discussion on lines 470-479 that with aging, upregulation of IL4 is implicated with cognitive decline.

- The procognitive effects of leukocyte-derived IL4 in young mice can be reconciled with the cognitive deficits observed following changes in the choroid plexus IL4:interferon-gamma ratio in aged mice. Aging is associated with accumulation of effector T-cells with the capacity to 'remember' previous insults and generate pro-resolving Th2 cytokines as a result. If Th2 cytokine signaling is not counterbalanced by Th1 cytokines, destructive autoimmune responses can result. Young animals likely exhibit appropriate counterregulation of Th2 cytokines, while the old animals in the study by Baruch et al (2013 PNAS) did not.

However, as noted by the Reviewer, the current studies were carried out in young adult mice. There are a few studies in aged rodents maintained on high-fat diet throughout the lifespan, but to our knowledge, none of those studies investigated IL4 or other Th2 cytokines. To avoid over-generalizing between the effects of obesity in young mice and the effects of aging, this section of the Discussion has been removed in the revised submission.

R2.Q2. The authors take the stance that such controversies are due to the lack of standardized physiological exclusion criteria in aging studies, however, this is unlikely as the NIH has been requiring and implementing the use of more complete procedural descriptions of diet, environment, and the overall health status and body condition of each animal in studies of aging.

- We agree that the new NIH criteria have increased the rigor and transparency of aging research. However, some studies referenced in the original submission were carried out before implementation of the new standards (e.g. Maher et al, 2005). Because there was no way to determine whether the study by Maher et al (2005) met current NIH standards for rigor and reproducibility in aging research, this reference was removed from the revised submission. Likewise, the reference in AD mice (Kawahara et al, 2012) involved combined intracortical injection of IL4 and IL13 (without including data on the effects of each cytokine individually). Because the approaches used in this reference lacked rigor, the citation has been removed from the revised submission.

- The rationale for using young adult mice was based on longitudinal studies in humans indicating that obesity at midlife increases dementia risk, irrespective of whether

individuals lost weight during the intervening decades (Whitmer et al, 2008). We introduce this potential 'critical period' in the Introduction of revised submission, but the overall discussion of aging/AD has been substantially reduced to better align with the timeline for these experiments.

R2.Q3. Also, note that studies of aging in rats have been able to identify very different populations of animals: those that age with little if any cognitive decline, and those that age with large deficits. This is in line with clinical data and perhaps should be considered in the interpretation of the data.

- We acknowledge that the discussion of aging in the original submission was problematic; as noted by the Reviewer, rats exhibit substantial inter-individual heterogeneity in cognitive decline. This heterogeneity is conserved in humans, but there is conflicting evidence from mice, especially inbred strains. In order to avoid overgeneralizing between species (mice and rats), developmental stages (young vs old) and conditions (obese vs lean), most references to aging have been removed from the revised submission.

R2.Q4. While it is clear why the authors chose to emphasize aging-related evidence (i.e., diabetes, obesity, sedentary lifestyle, etc...) in the rationale for the study as well as in the Discussion section, the work presented was accomplished in young adult mice. This point does not detract from the clarity, novelty, and thoughtful characterizations presented here, nevertheless, a caveat should be included to mention that for accurate alignment with clinical manifestations of the disease processes engaged in T2DM, experiments in aged animals should be considered.

- The rationale for discussing the aging literature was based on longitudinal studies indicating that obesity in adulthood increases dementia risk, irrespective of whether individuals lost weight during the intervening decades (Whitmer et al, 2008). We mention this potential 'critical period' in the Introduction section of the revised submission, but the overall discussion of aging/AD has been condensed to better align with the timeline for these experiments.

R2.Q5. While the authors provide significant and substantive evidence that SAT transplantation attenuates brain neuroinflammation in obesity through an IL4 pathway, they do not provide evidence for the manner in which beige adipocytes from the transplant interact with the recipient's lymphocytes in the brain. The Discussion may need to be expanded to clarify this and offer the reader a tentative mechanism for how peripheral perturbations impact CNS function at the BBB.

- Actually the probable site of action for IL4 in these studies was in CSF, as there was no difference in circulating IL4 (Fig.5C), and no evidence of IL4 gene expression among cells of the brain parenchyma (Supplemental Fig.3 in original submission; Supplemental Fig 7 in revised submission). We introduce the themes of blood-brain and blood-CSF interactions in the Introduction section of the revised submission. The Discussion section also includes additional text related to lymphocyte interactions with beige adipocytes (paragraph 3) and meningeal lymphocyte trafficking (paragraph 4).

R2.Q6. Please clarify what is meant by "earlier onset" on line 150, as it is not clear whether this addresses animal ages or suggests time is a variable in these flow cytometry experiments.

- We apologize and acknowledge that this was incorrect. The phrase "earlier onset" referred to the fact that Wt/HFD mice did not exhibit macrophage infiltration at 4wk (Fig.1F). Wt/HFD mice exhibited macrophage infiltration after 12wk in our recent

publication (Guo et al, 2020 JCI), and we interpreted this change as "earlier onset." To clarify, references to "earlier" or "accelerated" onset have been removed from the Results section.

R2.Q7. Also, please clarify line 209, as it not obvious from Figure 2E that the nTg/LFD show reductions in distance traveled, and no statistics are presented.

- The within-subjects effect of time was statistically significant and is reported in the Results section of the resubmitted version.

R2.Q8. Given the size and complexity of the study, the authors should briefly describe the number of animals or cohorts that went into these experiments.

- We apologize for this omission. It was not our intent to obscure the n-sizes and degrees of freedom were reported for each statistic in the original submission. However it is true that some methods of correction for heterogeneity of variance (e.g. Greenhouse-Geisser) reduce degrees of freedom, limiting its informative value for determining group sizes. The revised submission includes the exact n-sizes for panels in each Figure and full datasets are included as Supplemental Information. We also expanded discussion of design and methods in the Supplementary Methods document.

Reviewer #3:

This is a well written manuscript that represents an enormous body of work including mouse models, behavioral assays, cellular and molecular phenotyping, and electrophysiology. The findings highlight a novel role for beige fat in protecting against cognitive impairment related to obesity and provide a mechanism within the CNS that underlies these effects. As is true with any new finding, a large number of future experiments are now possible, but these are beyond the scope of this already very comprehensive study. Discussion of the following points would strengthen this already well done manuscript.

- We thank the Reviewer for their enthusiastic feedback.

R3.Q1. Throughout the text, the authors ascribe the effects to beige fat. While this may well be true, it seems there are at least two possibilities: (a) beige fat itself is neuroprotective or (b) in the absence of beige fat, subcutaneous fat acquires more visceral-like properties which would be deleterious for cognition. The ability of wild-type SAT transplants to correct the phenotype seems to argue more for possibility (a).

- The Reviewer is correct in noting that there are multiple potential mechanisms for these effects. However, we would like to respectfully emphasize that possibilities (a) and (b) are not mutually exclusive. As noted by the Reviewer, possibility (a) is supported by cognitive rescue and reversal of microglial activation following SAT transplantation from nontransgenic donors, but not from transgenic donors lacking beige fat (Figs.3-4). However, in the revised manuscript, possibility (b) is supported by newly added gene expression data (Supplementary Fig.3), which suggest that SAT acquires VAT-like pro-inflammatory signatures in beige fat knockout mice with dietary obesity.

Acquisition of VAT-like features was only detected in beige fat knockout mice with dietary obesity, not in nontransgenic mice with dietary obesity or in lean mice lacking beige fat (Supplementary Fig.3). Because nontransgenic mice with dietary obesity exhibit beige adipose-dependent cognitive rescue following SAT transplantation (e.g. Fig.3), it is likely that beige adipocytes in transplanted SAT are neuroprotective (scenario a), and that this neuroprotection involves IL4 production by recipient-derived lymphocytes.

R3.Q2. However, wild-type SAT from room temperature housed mice has very few beige adipocytes, as these cells are typically induced in response to cold exposure or beta-adrenergic agonism. Have the authors considered testing the effects of SAT transplants from cold-exposed mice? It is possible that these transplants could have even stronger beneficial effects though it is unclear how long this phenotype might persist upon transplantation.

- We agree that cold exposure and beta-adrenergic agonists stimulate beiging in SAT, and that beige adipocytes are extremely rare at thermoneutrality (30C for mice). However, the animal colony was maintained at 22C, and multiple groups have demonstrated there is a sizable population of beige adipocytes in SAT at this temperature (e.g. Chi et al, 2018; Cohen et al, 2014).
- The revised submission also includes new data that reflect the persistence of beige adipocytes in the transplant, as reflected by UCP1 immunoreactivity and the presence of multilocular adipocytes (Supplementary Fig.5A-B). qPCR analysis of thermogenic genes was also performed for the transplanted SAT (Supplementary Fig.5C). These data indicate that transplanted SAT retains features that are consistent with donor genotype.

R3.Q3. Aerobic exercise has also been shown to attenuate cognitive decline. Since exercise activates beige fat, the authors could consider in the future testing whether beige fat is required for the cognitive benefits linked to exercise. A similar type of transplant model has been employed to assess other benefits linked to beige fat (<https://pubmed.ncbi.nlm.nih.gov/26050668/>).

- We agree that it would be very interesting to investigate these effects using a beige fat gain-of-function model, such as exercise or cold exposure. Those experiments will definitely be pursued in the future.

R3.Q4. It would be nice if the authors speculated on possible mechanisms by which beige fat might mediate these changes in the brain. Do the authors believe there is a humoral factor that signals to the brain or a neuronal circuit?

- This is certainly an important question. We believe that the data in this manuscript suggest that there is a humoral factor. This interpretation is based on the data in Fig.5B, which shows increased CSF IL4 with SAT transplantation from a Wt donor, but not from donors lacking beige fat. There was no change in circulating IL4 (Fig.5C), suggesting that immune cells are signaling via lymphatic pathways, or by trafficking between adipose tissue and the CNS. This issue was indirectly addressed by the data in Fig.6, which indicate that immune cell trafficking between transplanted SAT and the CNS is unlikely (Fig.6C), and that recipient-derived lymphocytes are required for the beneficial effects of SAT transplantation (Fig.6D-G).
- We considered the possibility of a retrograde trans-synaptic signaling mechanism linking sympathetic innervation of SAT with these results. The revised submission includes immunohistochemical visualization of tyrosine hydroxylase-stained fibers in SAT. There was no evidence of genotype differences (Supplementary Fig.1E), and no clear evidence of re-innervation in transplanted SAT (Supplementary Fig.5C). To our knowledge, no tracing studies have identified a polysynaptic pathway linking hippocampal neurons with SAT. These data, and prior work on sympathetic innervation of SAT, make the retrograde trans-synaptic hypothesis somewhat unlikely.

R3.Q5. The authors focus on microglia in this study, but have they considered a role for other cells that might respond to IL4, like astrocytes?

- We considered the potential role of astrocytes as a source of IL4, but that was not supported by the data (Supplementary Fig.7). However, it would be fascinating to delve deeper into astrocyte-microglia interactions in the context of obesity, and we are currently breeding mice for this purpose. In the Discussion section of the revised manuscript (pasted below), we outline the basis for our interpretation of the role of microglia in these effects, and we acknowledge that this interpretation is based on indirect evidence. While considerable work remains to be done, the current report clearly demonstrates the existence of an immunoregulatory relationship between beige adipocytes and the CNS.

"The IL4 receptor-alpha chain (IL4Ra) is expressed by neurons, microglia, and astrocytes (Nolan et al, 2005; Fenn et al, 2014; Barna et al, 2001). However, downstream signaling differs between microglia and other CNS cell types. In microglia, IL4Ra activation results in dimerization with the common gamma-chain of another IL4R, forming the type 1 signaling complex (Wills-Karp and Finkelman, 2008). In neurons and astrocytes, activation of IL4Ra leads to dimerization with interleukin-13 receptor alpha1 (IL13Ra1), forming the type 2 signaling complex (Barna et al, 2001). The cellular consequences of both receptor complexes are anti-inflammatory, but the signaling cascades recruited by type 1 or type 2 signaling are distinct. The type 1 receptor complex recruits insulin receptor substrate (IRS) 1/2, while both receptor complexes activate JAK1/STAT6 (Wills-Karp and Finkelman, 2008). Signaling downstream of IRS1/2 activates kinases that enhance LTP, including Pi3K/AKT and ERK1/2 (Malenka and Bear, 2004). Activation of JAK1/STAT6 has not been directly linked with synaptic plasticity, but other members of the JAK/STAT signaling pathway are selectively required for hippocampal long-term depression, and do not alter LTP (Nicolas et al, 2012). In this report we observed rapid reinstatement of LTP in slices treated with minocycline, and rapid elimination of protection in SAT transplant recipients following incubation with an IL4 scavenging antibody. While temporal kinetics do not demonstrate cell type specificity, microglia represent a more likely cellular substrate than astrocytes given the different receptor complexes recruited by IL4 in the two cell types and the distinct effects of downstream signaling cascades on LTP and LTD."

R3.Q6. In the behavioral studies, the authors made efforts to account for factors like visual acuity and motor abilities, but not for anxiety levels. Is there any concern that there could be a difference in these groups?

- Differences in anxiety-like behavior have been reported by some groups, but there is also an extensive literature in rodents on the anxiolytic effects of palatable diets (e.g. 'comfort food'). In our hands, mice with diet-induced obesity do not exhibit basal anxiety, as reflected by the absence of group differences in exploration of the empty arena during the initial phase of novel object testing, which is essentially identical to the open field. There were also no differences in exploration of the center of the empty arena. These data are included under relevant sections of the Results section in the revised manuscript.

Reviewers' Comments:

Reviewer #1:

Remarks to the Author:

My previous concerns were satisfactorily addressed, and I endorse this article to be published with no further revisions. It will be an excellent addition to the available literature.

Reviewer #2:

Remarks to the Author:

The authors have addressed all initial comments and points of clarification and now provide a refocused manuscript that is more complete, and that offers interpretations well-aligned with the experiments presented. The results are clearly presented and while it was not suggested to remove all tentative associations between the data obtained and aging or AD processes (brain or periphery), the current refocus of the work is welcome. The new data included on genes associated with being also help to clarify the features of the transplantation experiment that are important for interpretation of the main results.

Reviewer #3:

Remarks to the Author:

With their rebuttal and additional data, the authors have now addressed all of the concerns raised in my initial review. Their manuscript is well written with rigorous and compelling data.